# Asymmetric division events promote variability in cell cycle duration in animal cells and *Escherichia coli*

Ulrich Berge [1,2], Daria Bochenek[1,2], Ralf Schnabel[3], Arne Wehling[4], Timm Schroeder[4], Tanja Stadler[4,5] & Ruth Kroschewski[1]

Asymmetric cell division is a major mechanism generating cell diversity. As cell cycle duration varies among cells in mammalian tissue culture cells, we asked whether their division asymmetry contributes to this variability. We identify among sibling cells an outlier using hierarchical clustering on cell cycle durations of granddaughter cells obtained by lineage tracking of single histone2B-labelled MDCKs. Remarkably, divisions involving outlier cells are not uniformly distributed in lineages, as shown by permutation tests, but appear to emerge from asymmetric divisions taking place at non-stochastic levels: a parent cell influences with 95% confidence and 0.5% error the unequal partitioning of the cell cycle duration in its two progenies. Upon ninein downregulation, this variability propagation is lost, and outlier frequency and variability in cell cycle durations in lineages is reduced. As external influences are not detectable, we propose that a cell-autonomous process, possibly involved in cell specialisation, determines cell cycle duration variability.

[1] Institute of Biochemistry, ETH Zurich, 8093 Zürich, Switzerland. [2] Molecular Life Science PhD Program, Life Science Zurich Graduate School, 8057 Zürich, Switzerland. [3] Institute of Genetics, Technical University Braunschweig, 38092 Braunschweig, Germany. [4] Department of Biosystems Science and Engineering, ETH Zurich, 4058 Basel, Switzerland. [5] Swiss Institute of Bioinformatics, 1015 Lausanne, Switzerland. Correspondence and requests for materials should be addressed to R.K. (email: Ruth.Kroschewski@bc.biol.ethz.ch)

Asymmetric cell divisions are present in all domains of life. As they cause heterogeneity among the two daughters of a dividing cell, they contribute to the generation of cell diversity and the emergence of different cell types. For example, asymmetric cell division is essential for the rejuvenation of at least some single-celled organisms and for the differentiation of some stem cells, where the two resulting daughter cells can differ in specific aspects linked to cell fate[1–4]. However, whether such diversifying asymmetric cell divisions concern only specific cell types or are a more general process, i.e., whether apparently symmetrically dividing cells establish some level of heterogeneity, is unclear. Remarkably, most animal cells form their mitotic spindle between two centrosomes that themselves contain centrioles of different age and structure[5,6]. Thus, at least in relation to centriole age most animal cells divide asymmetrically. Another feature of asymmetrically dividing cells is that one of the two daughter cells tends to take longer to re-divide than its sibling[4,7–9]. Thus, in some stem or cancer cells, cell cycle duration has already been followed as a marker of asymmetric cell divisions giving rise to cells with specific cell fates[10–12]. However, whether the variability of cell cycle duration, which is observed in many cell types, itself reflects some underlying asymmetry in cell division is not yet clear. Such variability can equally well be produced by internal processes, or noise, representing stochastic incidents or environmental factors. Therefore, a detailed study is needed to decipher the contribution of these aspects to the observed variability in apparently symmetrically dividing cells.

Even in highly controlled laboratory environments, such as microfluidic chambers, variations in cell cycle duration have been seen in the progeny of single bacteria and budding yeast[13,14]. Similar observations have been reported for mammalian tissue culture cells, where such variability has been known for many years[15–18]. The fact that such variability is observed ex vivo suggests that the observed variability in cell cycle durations relies on some cell-driven process. However, whether the variability of cell cycle durations observed during the clonal divisions of mammalian cells reflects some division asymmetry is unknown. To address this possibility, we designed long-term time-lapse microscopy assays and tested the resulting cell cycle duration data for biological noise, as well as internal and external contributions.

Studying the link between information propagation across cell divisions and the diversity of cell cycle duration, we identify relative outlier cells among four related granddaughter cells (3:1 motif) in lineage trees of diverse model systems. In lineages of si non-target-treated Madin Darby canine kidney (MDCK) cells we detect in a propagation test that outlier cells emerge from asymmetric divisions taking place at non-stochastic levels. This reveals an influence of a parent cell on its two progenies. However, such information propagation is not detectable at statistically significant levels in si ninein-treated MDCK cells and not due to detectable external influences. Further, the non-stochastic propagation of outlier cells is not restricted to small interfering RNA (siRNA)-treated MDCK cells but also detectable in the developing *Caenorhabditis elegans* zygote, differentiating mouse haematopoietic stem cells, and in *Escherichia coli* cultured at pH 7.5 but not in *E. coli* cultured at pH 6.0, and non-differentiating mouse embryonic stem cells. Moreover, in terms of cell cycle duration, abnormally frequent asymmetric divisions are present in all cell systems with non-stochastic propagation tests. These data suggest that the here identified outlier motif and its non-stochastic propagation report about cell-specialisation events. Together, we propose that a cell-autonomous process determines cell cycle duration variability as well.

## Results

**A frequent 3:L motif in lineages of single MDCK cells**. We aim to probe whether variability of cell cycle duration is linked to divisions of mammalian tissue culture cells in up to five times repeated experiments. For this, single MDCK cells expressing YFP-tagged histone2B (H2B-YFP, hereafter called MDCK cells) were transfected in parallel with non-targeting or ninein-targeting siRNA oligos and imaged for up to 85 h or 84 h, respectively, at multiple positions in the culture wells. Downregulation of ninein, a centrosomal protein, was shown to interfere with the characteristic partitioning of centrosomes in neuronal stem cells[19]. In the following, first only the results of the non-targeting siRNA treatment (si non-target) are presented, followed by those where ninein was downregulated (si ninein, Supplementary Figure 1). In both conditions, single cells divided forming colonies during imaging. In movies documenting colony formation, individual cells could be followed over time and their nuclei were tracked up to a maximum of seven generations (Supplementary Figures 2, 3 and Supplementary Movies 1, 2). From the 36 movies acquired of cells with the si non-target treatment in four independent experiments, we generated lineage trees and analysed the cell cycle durations (Fig. 1a, Supplementary Figure 2, see Methods and Pampaloni et al.[20]). In these 36 lineages, overall cell cycle durations were highly variable, ranging from 3.5 h to 48.3 h (Fig. 1b).

We wondered whether the variability in cell cycle duration of related cells originates from intrinsic processes segregating asymmetrically in divisions, our working hypothesis. To test this, we searched for re-occurring cell cycle motif(s) in the lineages. First, we computed all possible mother–daughter pairs, sibling cell pairs (daughter-daughter) and pairs of cells among granddaughter cells. We investigated whether cell cycle durations between the members of each of these pairs were rather the same or very different from each other (Fig. 1c), as also done in Sandler et al.[21]. The similarity of cell cycle durations is highest between daughter cell pairs followed by granddaughter cell pairs, and the least between mother–daughter cells, as reported[21]. As cells of the same generation rank are thus more likely to be similar, diversity motifs among these cells are particularly meaningful. Further, as diversity is among granddaughter cell pairs higher than among daughter cells, we looked whether these pairs formed special motifs. We therefore generated all granddaughter sets (set of the four cell cycle durations of the granddaughter cells of the same grandmother cell, e.g., Fig. 2a and Supplementary Figures 2, 4) and analysed whether these four cells are always very alike or show, relative to each other, reproducible diversity motifs in their cell cycle durations.

The theoretically most minimalist diversity motifs among the cell cycle durations of four cells are the following two: (i) a 3:1 motif, where one of the cells has a very different cell cycle duration (outlier cell) relative to the other 3 and (ii) a 2:2 motif, where the cell cycle durations are more similar within a pair than between the pairs (Fig. 2a). To capture in our lineages such motifs that are based on relative differences within a granddaughter set, we used the outlier detection method (ODM; Fig. 2b and Supplementary Figure 5). In case the four cells of a granddaughter set cannot be separated into two clusters with a given threshold, the method assigns the granddaughter set a flat motif. We focus in the following on the 3:1 diversity motif that introduces simply by its structure an in-balanced heterogeneity. This motif is also remarkable, as one of the two daughter cells that give rise to a 3:1-granddaughter set divided into an outlier and a non-outlier cell (Fig. 2a). The 3:1 motif occurs in two manifestations as follows: (i) the 3:L motif, one of the cells has a long cell cycle duration (L-cells) relative to the other three, and (ii) 3:S motif, one of the cells

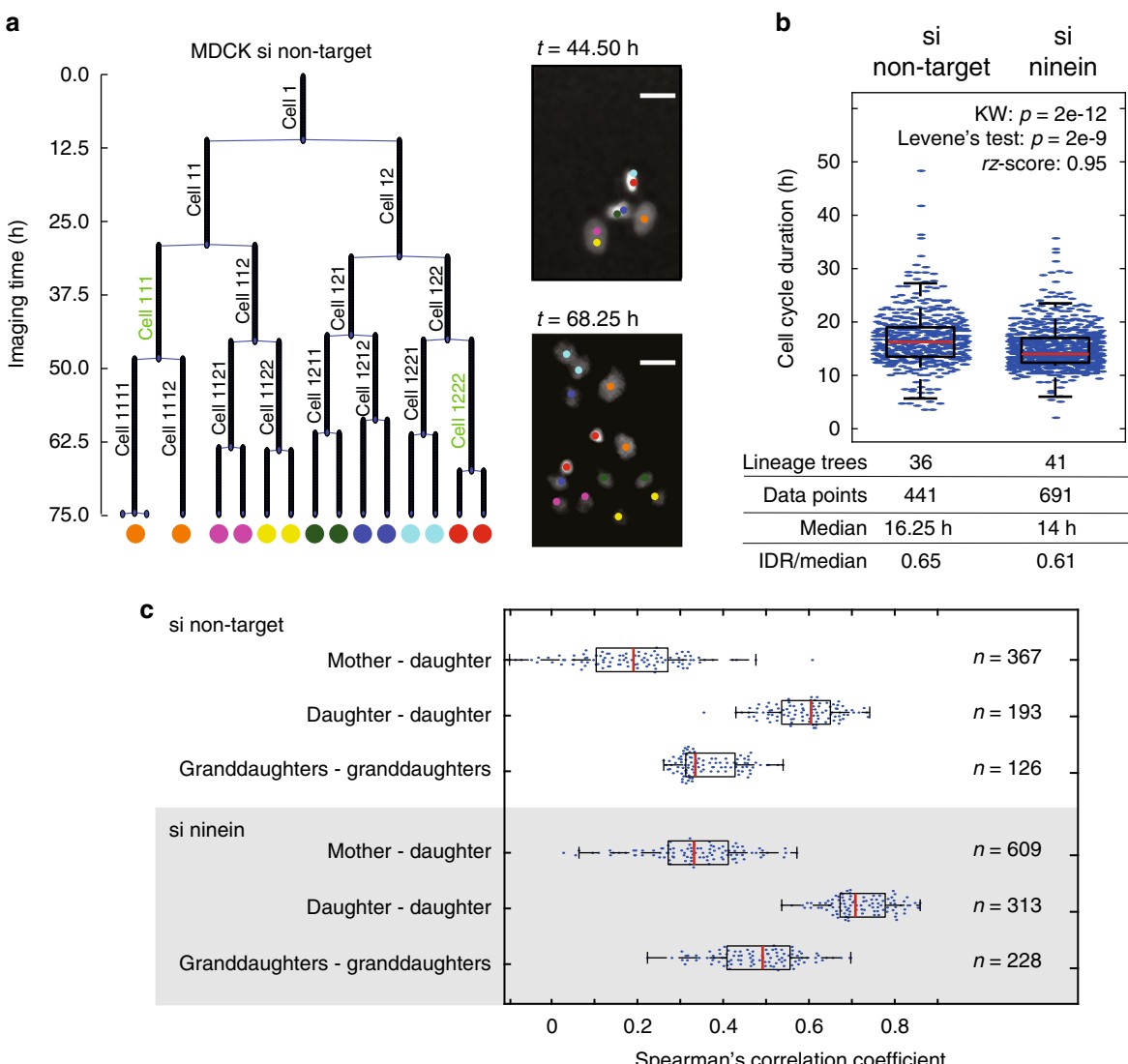

**Fig. 1** Variability of cell cycle duration is higher in si non-target MDCK cells than in si ninein-treated MDCK cells. **a** Left: tracked imaged lifetimes of cells with indicated cell identities (vertical terms), originating from a single cell over 75 h represented in a lineage tree (lineage identifier (ID) 10). Nodes depict cell divisions, edge length reports tracked imaged lifetimes of bijectively assigned cell identities (cell ID), which corresponds to cell cycle durations of these cells if the cell could be imaged over its entire lifetime. Green cell IDs indicate L-cells of the 3:L motif. Colour circles represent cell identifiers at imaging end. Right: fluorescence images of nuclei superimposed with colour circles at indicated time points of the lineage tree (e.g., blue and red cells—colour code tree imaging end). Bars 5 μm. **b** Distributions of cell cycle durations in si non-target and si ninein MDCK cells. Boxplot: central mark—median; edges of horizontal box—first and third quartiles; whiskers—1.5 times the interquartile range (IQR). Below: lineage trees—number of analysed lineage trees; data points (blue)—number of cell cycle durations; median—median cell cycle duration (red); IDR/median—interdecile range normalised to the median as quantifier for dispersion; p-values for Kruskall–Wallis (KW) and Levene's test; rz-score—robust z-score. **c** Spearman's correlation coefficients of cell cycle durations for all possible mother–daughter, daughter–daughter or granddaughter–granddaughter pairs of cells in si non-target and si ninein-treated cells. Boxplots (as in **b**) show the distributions of Spearman's correlation coefficients of 100 times randomly chosen cell pairs of the indicated types to compare the two conditions; individual data points after subsampling (blue circles). n number of mentioned cell pairs in the given data set that were subsampled

has a shorter cell cycle duration (S-cells) relative to the other three.

Specifically, each granddaughter set in the 36 si non-target lineages was ordered by magnitude of the cell cycle duration and then hierarchically clustered based on Mahalanobis distances (MDs) (Supplementary Figure 5). Afterwards, we calculated the dissimilarity index, defined as the distance ratio between the two most differing sub-clusters in a granddaughter set (Fig. 2c (left) and Supplementary Figure 5). The most meaningful value for this index is 2. A value above 2 indicates that the distance between the two most distant sub-clusters is larger than the sum of the smallest pairwise distances within the rest of the data. Therefore,

this value is used as a threshold to sort each granddaughter set into one of the motifs defined above, except for Fig. 2c, where a range of different thresholds is used to probe for the robustness of motifs. Using this method, we find that the 3:L motif is the dominating diversity motif over all tested thresholds (threshold 2: 3:L motif 55.5%, 3:S 9.5%, 2:2 motif 17.5%; flat motif 17.5%; Fig. 2c (vertical dashed line) and Supplementary Figure 4a). Furthermore, unlike the 2:2 motif, the 3:L motif is for several tested thresholds above (threshold 1.5, 2, 2.5, 3, 3.5, 4, 7) and the 3:S motif below (threshold 1.5, 2, 2.5, 3) the 2.5–97.5% confidence interval of a random scenario according the working hypothesis (Fig. 2c).

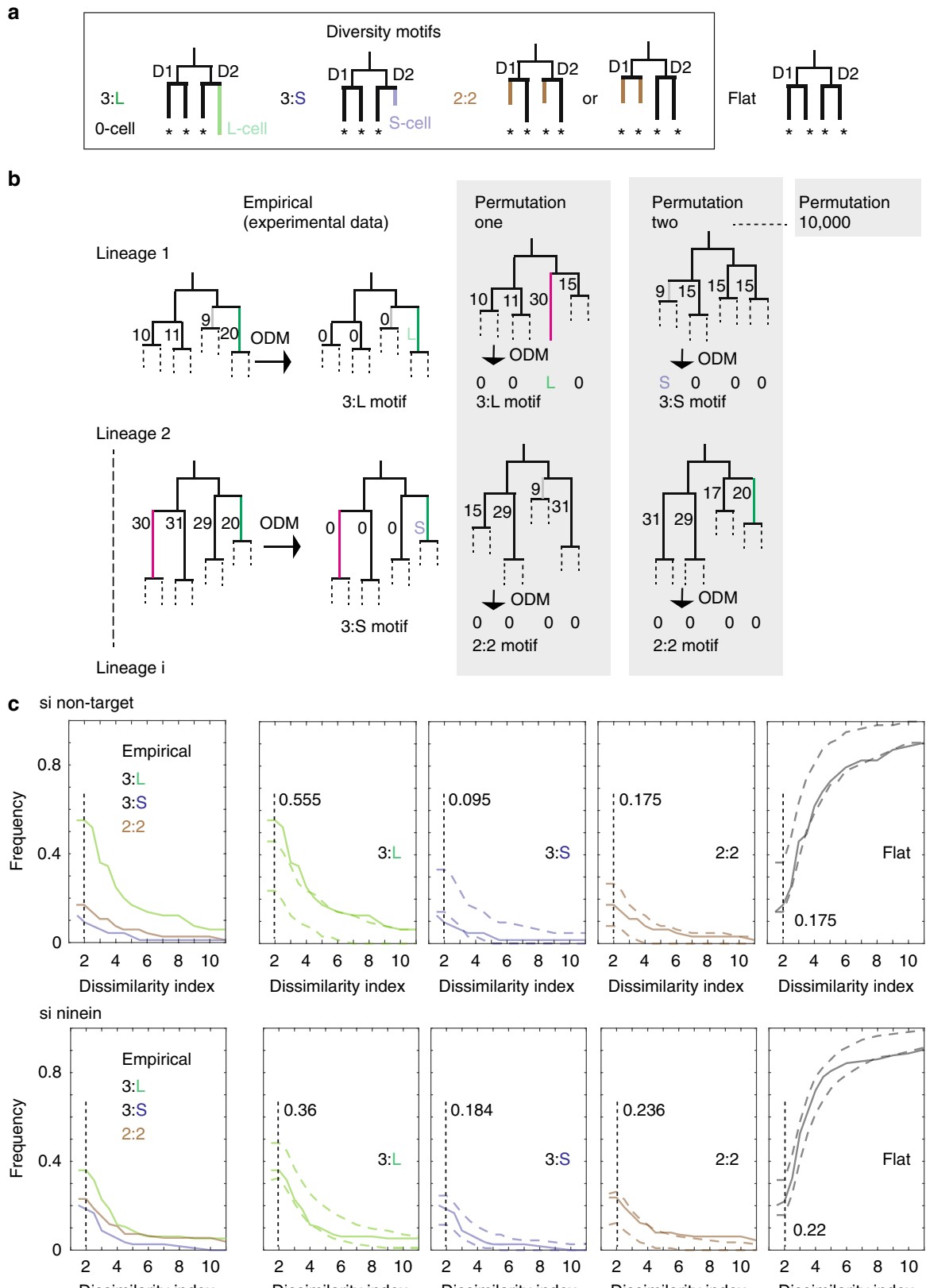

To understand the outstanding frequencies of the 3:1 motifs, we focus on the two kinds of these relative outlier cells, the L-cells and S-cells. We first asked whether the outlying cell cycle durations in the L- and S-cells might be due to aberrant mitotic lengths, but did not find support for this possibility. We therefore deduce that the interphase duration of the relative outlier cell is relevant for the striking cell cycle duration differences in 3:1 sets (Supplementary Figure 6).

**Fig. 2** Variability in cell cycle duration can be quantified by diversity motifs among granddaughter cells and experimentally altered. **a** Illustration of motifs and 0-, S- and L-cell classes. *0-cell; D1, D2 are the two daughter cells of the common mother cell. **b** Illustration of permutations, their effects on outlier positions and frequencies in lineage trees. Permuted were all cell cycle durations (without replacement) within each generation and across all lineages, maintaining the topology of the lineage trees. Here, (i) permutations of cell cycle durations only in generation 3 and (ii) on 2 lineages. Indicated numbers in the lineage tree parts represent arbitrary lengths of invented cell cycle durations, except if after employing the outlier detection method (ODM). Green cell (20 h), the pink cell (30 h) and the grey cell (9 h) of the empirical data set are assigned in permutations to different cell classes documenting the relativity of outlier cells. The outcome of the ODM for generation 3 is reported as motif and with 0-, S- and L-cells. **c** Frequency of the diversity motifs (3:L (green), 3: S (blue), 2:2 (brown)) and the flat motif in dependence on dissimilarity index thresholds in si non-target MDCK cells ($n = 63$ granddaughter sets) and below in si ninein MDCK cells ($n = 114$ granddaughter sets). Left plot: empirical frequencies of indicated motifs; right plots: bold line: empirical data, dashed lines: 2.5% (lower) to 97.5% (upper) confidence interval borders, respectively, estimated by permuting cell cycle durations per generation accounting for the working hypothesis and generation dependency (Supplementary Figure 16b). At threshold 2, vertical dashed lines with frequency numbers of empirical motifs

**Non-stochastic distribution of S- or L-cell divisions**. Next, we asked whether a cell-intrinsic process or noise contributes to the generation of the 3:1 motif. To this end, we probed whether a cell with an outlying cell cycle duration propagates some effect onto the cell cycle durations of its two daughter cells in the next generation, thus affecting the 3:1 motif frequency in the next generation. To statistically test this possibility, we analysed the positions of the relative outlier cells in the lineages comparing them with all the other cells, termed 0-cells (all cells that are neither L- nor S-cells) (Fig. 2a and Supplementary Figures 3a, 4c). The following propagation test reveals whether cell cycle durations of daughter cells are uniformly at random linked to mother cells, which we define here as a stochastic association, or if a non-uniformity exists, which we define as non-stochastic association between mother and the two daughter cells in terms of cell cycle duration.

For each S-, L-, or 0-class cell in a given generation, we assessed the cell classes of its two progeny cells in the next generation. With this, we obtained the empirical frequency of the nine possible division types, whereby S-, L- and 0-cells divide into (S/0), (L/0) or (0/0) pairs of daughter cells (Fig. 3a, b; other division types do not exist due to the nature of the 3:1 motif). Next, we tested whether the empirical frequencies of these division types are also very likely to occur after random permutations (non-parametric Monte Carlo permutations) of the cell cycle durations within each generation across all trees, or not (Figs. 2b, 3c and Supplementary Figure 7). In addition, we assessed the likelihood to obtain the empirical result by chance (type I error test; Fig. 3d, e and Supplementary Figures 7, 8).

In the 36 si non-target lineages ($n = 41$ granddaughter sets with a 3:1 motif, 70 divisions, Fig. 3b), the most frequent division type is the 0 to 0/0 type (61.4 %) and the second most frequent (22.9%) is the 0 to 0/L type, which is therefore the most frequent case of diversification observed here. The propagation test reveals that the division type frequencies are non-uniform at random, and that this result is likely due to two highly improbable division types (0 to S/0 and 0 to L/0) (Fig. 3f). In addition, we interpret the effect sizes of these shifts from random as biologically meaningful (Supplementary Figure 8 and Supplementary Table 2). Therefore, we conclude that the distribution of the division types in si non-target MDCK lineages holds a non-stochastic association. As the abnormally frequent division types are de novo generating outlier cells, our statistical results suggest that some cells are autonomously enabled to divide asymmetrically, i.e., to generate two daughters with different abilities to proceed through the cell cycle. To investigate further the possibility that diversification is cell autonomous, we tested whether modulating the frequency of asymmetric cell divisions alters the pattern of outlier generation.

**Si ninein erases the statistical influence of parent cells**. We thus asked whether the statistical result in si non-target MDCK

lineages outlier division types occur at non-stochastic frequencies reflects some biological reality or is a false positive result. For this we tested statistically whether a treatment that could perturb propagation fidelity across divisions abolishes the statistical non-stochasticity. Segregation of cell fate and cell-fate determinants are strongly correlated with the inheritance of either the old or the young centrosome[3,19,22–24]. At least in mouse neuronal progenitor cells, downregulation of the centrosomal protein ninein disturbed this correlation[9,19,25–27]. Therefore, ninein downregulation seems to affect centrosome inheritance and we asked whether it also affects the above established non-stochastic cell cycle duration propagation. To this end we tested whether ninein downregulation affected cell cycle durations and their distribution in the lineage trees (41 lineages in 5 independent experiments, 691 cell cycle durations; Fig. 1b, c, Supplementary Figures 1, 3b, 4a, b, 6, 8, 9 and Supplementary Tables 1, 2) compared with si non-target lineages. Employing our statistical propagation test of relative outlier cells on the si ninein data, we find that the statistical significance is erased (Fig. 4a and Supplementary Figure 9). Thus, in si ninein lineages, parent cells do not have a statistically detectable influence on the diversity of the cell cycle durations of their progeny cells, when focusing on relative outlier cells. This suggests that the significant propagation result obtained for the si non-target lineages is not a false positive one.

**Absence of detectable external influences**. There is the possibility that extrinsic rather than cell-intrinsic influences cause the difference between the two experimental conditions. Thus, we probed whether the statistical propagation difference between si non-target and si ninein lineages might be caused by external influences. For this we assessed colony shapes and colony-shape changes over time. We do not find differences between the si non-target and si ninein conditions (Fig. 4c, d). We also do not find relative outlier cells more frequently in a specific environment within the developing colonies (e.g., Fig. 4b). In both conditions, the diversity motifs are neither obviously enriched nor reduced in specific generations in trees (Supplementary Figure 4b). L- and S-cells are also not more motile than any of the other cell categories and do not localise more frequently at the periphery than in the centre of the colonies compared with any other cell (Fig. 4e, f and Supplementary Figure 10). As we did not detect environmental effects that correlated with the emergence of relative outlier cells, we assume that cells of the same generation are similarly influenced by their environment in both conditions. As the cells of both conditions were imaged in parallel, we conclude that the environmental influences on individual cells were not detectably different between both conditions and thus cannot explain the loss of the statistically documented influence a mother cell has on its daughter cells with respect to cell cycle duration. Consequently, si non-target cells, but not si ninein cells, propagate non-stochastically cell cycle duration diversity to the next generation

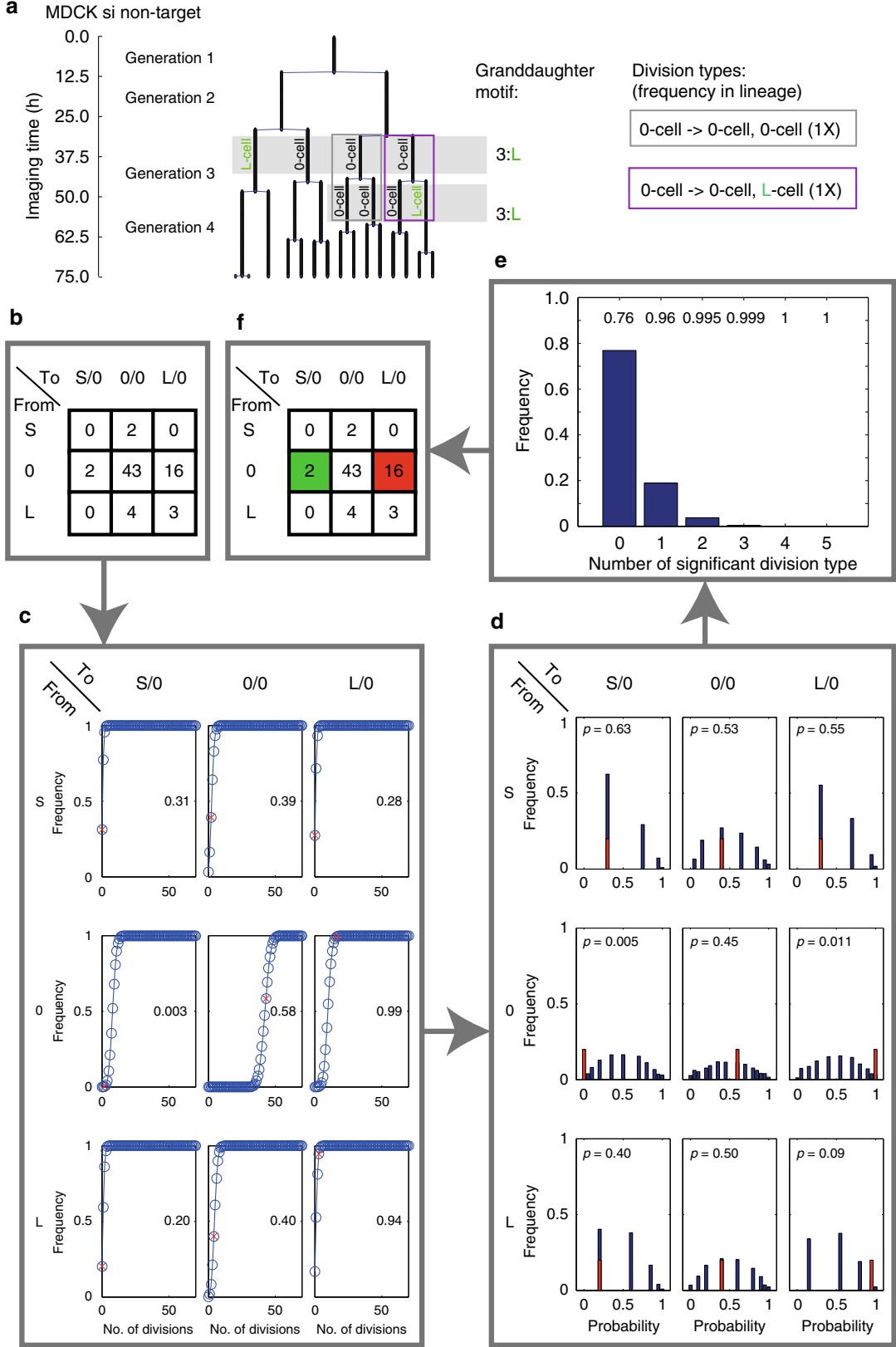

seemingly due to cell-intrinsic and not external differences. Consistently, the median cell cycle duration (si ninein: 14 h, si non-target: 16.2 h) and variance (Fig. 1a) are reduced in si ninein lineages compared with si non-target lineages. The cell cycle durations of even the mother–daughter pairs but also

daughter–daughter and granddaughter pairs are more similar compared with the si non-target condition (Fig. 1c). In addition, in si ninein-treated cells, both the 3:L and the 3:S diversity motifs occur at stochastic frequencies (Fig. 2c). Hence, the data are consistent with the statement that the differences in cell cycle

**Fig. 3** Statistical test detecting non-stochastic propagation of cell cycle duration variability across divisions. Division types involving outlier cells are uniformly at random in si non-target MDCK lineage trees (null hypothesis). Test rejection reveals with ≥95% confidence (two-sided) that mother cell classes influence non-stochastically the two daughter cell classes. 0-cell, all cells of granddaughter sets, which do not qualify as S- or L-cell (threshold 2). **a** Illustration of division types involving outlier cells in the lineage tree of Fig. 1a. Each filled grey square combines cells of one 3:L motif. **b** Empirical counts of division types of all si non-target lineages (dissimilarity index 2). "From" indicates the mother cell class, "to" the two daughter cell classes. Note: a cell dividing into S/S, L/L or S/L daughters is impossible. **c** For each division type, a cumulative distribution function (CDF) was estimated by 10,000 times permuting empirical cell cycle duration data without replacement over all lineages within the same generation (blue) and the empirical probabilities (red x; number in each field) calculated. **d** Estimation of the probability distribution function of obtaining any possible occurrence value per division type (blue bars). The procedure in **b** and **c** was repeated 10,000 times, each time substituting the empirical probability value per division type with a value from one of the randomisations (**c**). The empirical probability (**c**) of each division type is represented as a red bar of arbitrary height in the histogram. If this empirical probability ($p$) was outside the 95% confidence interval (two-sided) for a given division type, this division type was significant. **e** Power of the test. Frequency (blue bars) of significantly different division types based on permutations in **d**. Numbers in the graph represent the cumulative probability of having at least $x$ significant division types (in **d**; $x$ between 0 and 5). Here, two or more significant division types have a probability of 0.005. **f** Representation of the propagation test result: Only if the 5% probability criteria in **d** and **e** were met, matrix fields are colour-coded: red—most likely overrepresented, green—most likely underrepresented division types

durations between si non-target and si ninein cells are most likely due to a si ninein-sensitive, cell-autonomous process taking place in the si non-target condition.

**Modifiable propagation of outlier cell cycle durations**. Next, we probed whether the non-stochastic propagation of cell cycle duration variability of the mother cell to its two daughters only can be detected in siRNA-treated MDCK cells. We therefore analysed several other cell systems. Data of published *E. coli* lineages grown in environments with pH 7.5 or at pH 6.0 were statistically analysed for propagation of cell cycle duration variability, as before[28]. Remarkably, in the pH 6.0 data set (81 granddaughter sets), non-stochastic mother–daughter linkage is present probably due to the improbable frequencies of two division types (0 to S/0 and 0 to 0/0), whereas it is absent in the pH 7.5 data set (81 granddaughter sets; Fig. 5b, Supplementary Figures 11, 12; estimated robust z-scores (rz-scores) in Supplementary Table 2). At pH 6.0 but not at pH 7.5, the *E. coli* cells inheriting the old pole divided significantly slower than cells inheriting the young pole[28] and old pole cells had both reduced growth rates and subsequent divisions[13], indicating that the 3:1 motif might be linked to cell specialisation.

We also analysed mouse stem cell lineages obtained from imaging of in vitro cultures. Embryonic stem cells dividing in non-differentiation medium do not propagate outlier cell cycle diversity across divisions (78 granddaughter sets; Fig. 5c and Supplementary Figure 13)[29]. However, mouse adult haematopoietic stem cells under differentiation conditions[30] significantly do, probably due to the non-stochastic frequencies of two division types (0 to 0/0 and 0 to L/0) (332 granddaughter sets Fig. 5d and Supplementary Figure 14; estimated rz-scores in Supplementary Table 2). Thus, it seems that also in these non-siRNA-treated mammalian cells, cell cycle diversity captured by the 3:1 motif is non-stochastically propagated and linked to cell specialisation.

Finally, we examined a system where cell fate is known to segregate asymmetrically, the lineage of the *C. elegans* zygote (P0) (Fig. 5e, f and Supplementary Figures 15, 16a, f). In this lineage, four consecutive asymmetric divisions occur, each generating a germline precursor cell called $P_1$, $P_2$, $P_3$ and $P_4$, respectively. These germline precursors take longer to proceed through the cell cycle than their somatic siblings[31]. However, whether granddaughter sets in such lineages show any specific type of motif is unknown. Thus, we asked whether the long cell cycle durations classified the germline precursor cells as L-cells in a 3:L motif. We find that outlier cell cycle diversity across divisions is propagated in a non-stochastic manner (49 granddaughter sets; Fig. 5e and Supplementary Figure 15). This time, four division types can be the likely cause for the non-stochasticity (0 to 0/0, 0 to L/0, L to 0/

0 and L to L/0) (estimated rz-scores in Table 2). Importantly, L-cells are occupying the germline precursor positions in 95.2% (20 of 21 found up to G5) of all analysed divisions between $P_1$ and $P_4$ (Fig. 5f). Thus, these outlying long cell cycle durations are nearly always associated with a specific cellular identity. Interestingly, these L-cells are the cells with the highest proliferation and fate potential, as they generate the germline.

Together, due to (i) the non-stochastic positions of S- and L-outlier cells also in *E. coli*, mouse haematopoietic stem cells and *C. elegans* lineages (Supplementary Figure 16a), and (ii) the modifiable significance in the propagation test by external pH manipulation in the *E. coli* system, we can exclude that the results obtained for siRNA-treated MDCK cells are caused by siRNA treatment or are a unique property of the MDCK cell system. These additional cell system results point further to a biological relevance for the L-, S- and 0-cell classes in diverse cell systems. These results together with the non-detectable external influences on relative outlier cells and the stochasticity of the propagation test in si ninein MDCK cells, let us to argue strongly that si non-target-treated MDCK cells generate cell-autonomously variability in the cell cycle duration of their progeny as they divide. These results further suggest that other cell systems propagate also cell-autonomously cell cycle duration variability across divisions. Finally, the 3:1 motif appears as a robust signature of that diversification process and seems to be conserved in animals and *E. coli*.

**Discussion**
Here we identified parameters promoting the emergence of cell cycle duration variability in several clonally grown cell systems. Due to the single cell tracking and comparative analyses, our study extends beyond previous modelling-based reports[15–18,21]. Our data establish that cell cycle duration variability is not only influenced by extracellular and noise components but also expresses a cell-autonomous non-stochasticity dictated by the mother cell. This non-stochasticity manifests itself in the here discovered 3:1 diversity motif, in which one cell has an outlying cell cycle duration relative to those of the three other closest related cells of the same generation.

The 3:1 motif is the result of a division asymmetry of a daughter cell that gives rise to an outlier and a non-outlier cell of the motif (Fig. 2a). Consistently, a more frequent and stronger cell cycle duration heterogeneity among daughter–daughter pairs is indeed detectable in all generations in the si non-target condition (Supplementary Figure 17), where the frequencies of outlier division asymmetries are at non-stochastic levels (Fig. 3). This heterogeneity is less pronounced in the si ninein condition. Further, si ninein treatment of MDCK cells (i) decreased the

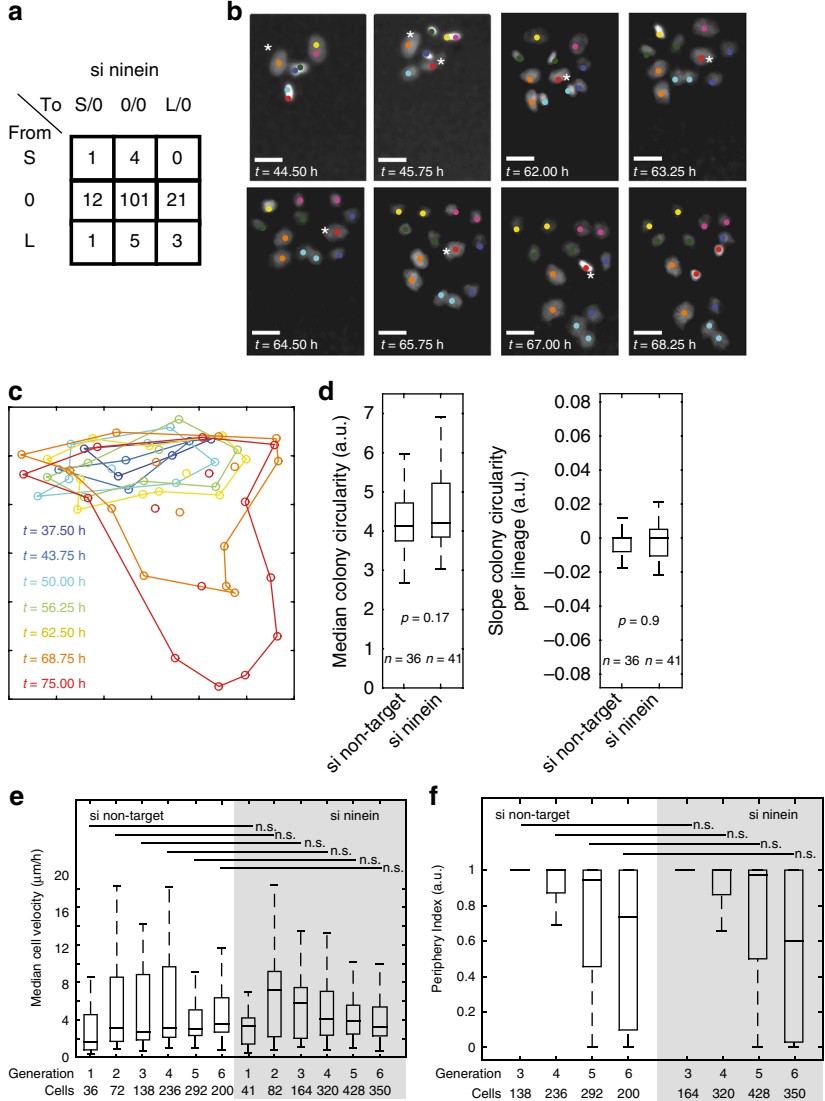

**Fig. 4** Biological controls for the non-stochasticity in the propagation of variability of cell cycle duration in si non-target MDCK. **a** Representation of the results of the propagation test for MDCK si ninein (as Fig. 3f). **b** Cell positions in the forming colony at selected time points of the si non-target lineage presented in Fig. 1a including colour code. Asterisks mark L-cells (orange and red labelled nuclei). Bar 5 μm. **c** Illustration of the cell colony shape in **b** based on the position of the nuclear centroids at seven time points (superimposed and colour-coded). Analogous circumference lines are the basis for determining the colony circularity and periphery indices. **d** Approximation of the gross colony shape and its change over time in colonies of si non-target and si ninein cells. Boxplot: central mark—median; edges of horizontal box—first and third quartiles; whiskers—1.5 times IQR. *n* number of colonies analysed. Left: comparison of median circularities for each colony during its imaging time. (KW test, *p*-value) Right: to document the gross change of colony shapes over time per condition, the circularity for each colony was fitted to a linear model as a function of time (robust linear regression was used to compute slopes. Comparison of slope distributions was tested according to Kruskal–Wallis (*p*-value)). **e** The median cell velocities change over generations per siRNA-treated condition but are not different between si non-target and si ninein cells of the same generation. Data of si ninein cells on grey background. Boxplot as in **d**. **f** The periphery index, expressing the ratio of peripheral vs. central colony position of each cell during its lifetime, changes over generations but is not different between si non-target and si ninein cells. Data of si ninein cells on grey background. Boxplot as in **d**; n.s., not significant at the significance level of 0.05

outstanding frequency of the 3:L motif, (ii) increased the improbable frequency of the 3:S motif and (iii) erased the propagation of cell cycle duration variability that is present in si non-target MDCK cells, suggesting ninein to be involved in generating cell cycle duration diversity in si non-target MDCK (Figs. 2c, 3, 4a, 5a). However, *E. coli* does not express ninein but can propagate outlying cell cycle durations at non-stochastic levels, suggesting that the molecular causes for the 3:1 motif differ.

The non-stochastic propagation of cell cycle duration variability (i.e., a non-stochastic propagation matrix composed of relative outlier cells) could represent a useful reporter for cell

specialisation events occurring in cellular diversification processes. At pH 6, an external stress to *E. coli* cells, but not at pH 7.5, non-stochastic propagation was detectable (Fig. 5b). This suggests that the tested cell classes (L-, S-, 0-cells) are linked to a stress response possibly allowing individual members of the population to cell-autonomously regulate the cell cycle duration of their progeny to support the survival of the species. This seems to be achieved by an increase of 0 to 0/0 and reduction of 0 to 0/S divisions. We identified in *C. elegans* lineages the germline precursor cells as L-cells (Fig. 5f). These cells are prospective founder cells of new organisms, documenting their high division and fate

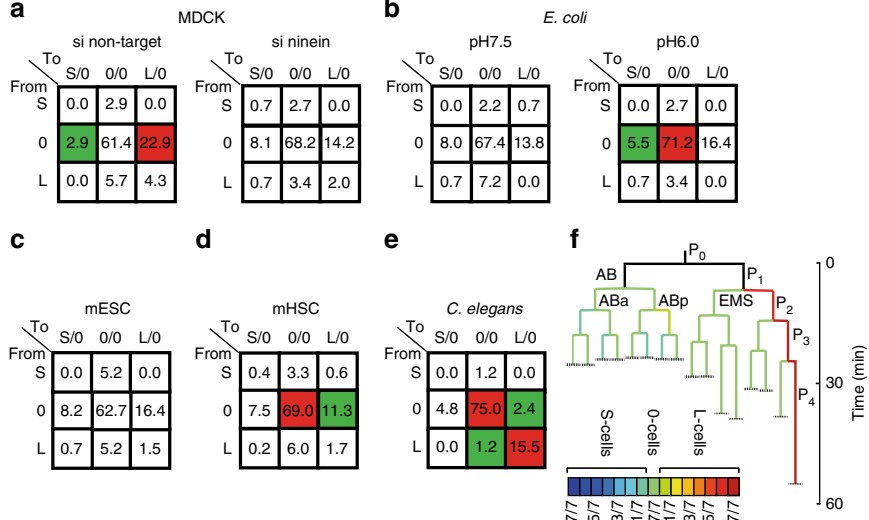

**Fig. 5** Propagation of cell cycle duration variability is present in diverse not siRNA-treated cell systems revealing a physiological relevance for outlier cells. Frequencies of the division types and result of the propagation tests for **a** MDCK si non-target and si ninein, **b** *E. coli* grown at pH 7.5 and pH 6.0, **c** non-differentiating mouse embryonic stem cells (mESC), **d** differentiating mouse haematopoietic stem cells (mHSC) and **e** developing *C. elegans* embryos. Colour code as Fig. 3. **a–e** Colour-coded matrix fields: red—most likely overrepresented, green—most likely underrepresented division types. Absolute number of divisions per model system in Supplementary Table 1. Matrices with absolute values in Figs. 3f, 4a and Supplementary Figures 9, 11–15. **f** An averaged lineage tree based on median cell cycle durations of seven *C. elegans* developments up to generation 5 is shown with the *C. elegans* typical cell names. Line colours indicate the frequencies of 0-cells (green), L-cells (green to red) and S-cells (green to blue) as defined in the left heatmap referring to the empirical frequency of individual cell types within the seven *C. elegans* lineage trees; black line, if granddaughter sets were not present

potential. Finally, whereas embryonic stem cells in non-differentiation medium—remaining in the self-renewal mode, where no differentiation takes place—did not propagate outlier cell cycle durations above random levels (Fig. 5c), differentiating haematopoietic stem cells did propagate cell cycle variability at non-stochastic levels (Fig. 5d).

Remarkably, in different cell systems the cause for the non-stochasticity in the propagation test was likely due to different division types, but always at least one of the likely critical division types was an asymmetric one (Fig. 5). Thus, although these examples all point to the 3:1 motif being involved in some kind of cell specialisation processes, the biological functions and underlying molecular mechanisms seem to differ. For future studies, establishing correlations between the relative outlier cells and cell size, cell differentiation markers, ageing reporters (aggregated proteins, DNA damage markers), the cilium or division potential might be promising first steps to identify the underlying molecular mechanism(s) of the cell-autonomous determinism identified here and to investigate its function, as it may be linked to cell specialisation decisions.

## Methods

**MDCK cell culture.** Low-passage (<P20) single MDCK cells (source: since decades kept in the I. Mellman laboratory (when at Yale University, USA); not listed in the ICLAC database v8.0, authenticated by 16SRNA fragment amplification; mycoplasma-free before and after the experiments) were seeded into 6 cm dishes and transfected with a plasmid encoding H2B-YFP[32] using Effectene (Qiagen). Cells were kept in selection medium (0.6 mg/ml G418 (Invitrogen) for 1 week. Cells were transfected with JetPrime reagent (Polyplus Transfection, Illkirch, Switzerland) and 0.5 nmol ON-TARGETplus Non-Targeting Pool of oligonucleotides (cat-number: D-001810–10–20: Dharmacon) are in the other parts of the manuscript named MDCK cells or indicated with si non-target so that they can be compared with cells, in which ninein was downregulated using a pool (1:1:1:1, total amount: 0.5 nmol) of four oligonucleotides (sense strands):

siRNA1: 5′-r(GAAUAUUGAUGGAGAGAUA)dTdT-3′
siRNA2: 5′-r(CAGAGAAGCUGGCCGAAUA)dTdT-3′
siRNA3: 5′-r(CAAGAGAACAUGAAGCAAA)dTdT-3′
siRNA4: 5′-r(GAGCAGCAGUGUAGGGUAU)dTdT-3′

all targeting open reading frame regions of canis lupus ninein (si ninein cells). After a double pulse of transfection, the cells were sparsely seeded into eight-well Ibidi chambers (1000 cells per well) with MDCK medium (MEM (Sigma), 20% fetal calf serum with 2 mM glutamine (Invitrogen), 100 U/ml penicillin and 100 µg/ml streptomycin, and 80 µg/ml insulin (Sigma-Aldrich, Spruce, USA)). For knockdown efficiencies, see Supplementary Figure 1.

**MDCK widefield imaging.** Immediately after attachment, isolated single, sparsely seeded, low H2B-YFP-expressing cells were selected and multi-position imaging was performed with an inverted Zeiss 200 M microscope ($\Delta t = 15$ min (experiments 1 and 2) or 20 min (experiment 3 to 5), total $t = 84$ h, fluorescein isothiocyanate channel and phase contrast every fourth time point) with a ×20 0.4 NA Korr Ph2 LD Plan NeoFluar objective in a humidified incubation chamber (5% $CO_2$/air, 37 °C) in up to 120 positions (MetaMorph Version 7.5.6). To avoid disturbances during the long-term imaging yet to grant excess nutrition and liquid, the culture dish was filled to the maximum with medium. Imaging was performed in ScopeM, ETHZ.

**Antibodies and chemicals.** The following antibodies were used: anti-γ-tubulin (Sigma (T6557) ms 1:5000), anti-IgG secondary rabbit (Invitrogen (A11034) gt 1:500) and anti-IgG mouse (Invitrogen (A21236) gt 1:500) antibodies labelled with Alexa Fluor dyes. Rabbit anti-Ninein (1:1000) was a kind gift from Michel Bornens (Institute Curie, Paris, France[33]).

**Immunofluorescence.** Briefly, the cells were fixed in cold (−20 °C) methanol for 6 min at −20 °C, detergent treated and blocked in 2% bovine serum albumin, 0.05% Tween-20 in phosphate-buffered saline (blocking buffer) and incubated in primary antibodies for 60 min followed by secondary antibodies for 30 min. All antibodies were diluted in blocking buffer. Nuclei were stained with Hoechst 33258. Coverslips were mounted on glass slides in Mowiol.

**Confocal microscopy.** Fluorescent images of fixed cells were recorded using Leica SP2 AOBS confocal microscope (Leica Microsystems). High-resolution XY-stacks of the cells (voxel size, 0.116 × 0.116 × 0.122 µm) were obtained with an oil-immersion objective (×60, 1.4 NA, HCX Plan-Apo).

**Real-time reverse transcription PCR.** For in vitro transcription, 2 µg of total RNA (isolated using NucleoSpin RNAII Macherey-Nagel) was treated with DNAse I (Promega), then reverse transcribed, using 200U M-MuLV RT and 80 ng random primers (Invitrogen), according to the manufacturer's instructions. All samples within an experiment were reverse transcribed at the same time, the resulting cDNA treated with 2U RNaseH (BioLabs), diluted 1:10 in nuclease-free water and stored in aliquots at 20 °C until used. Subsequently, 2 µl of transcribed RNA was

used in a quantitative real-time PCR reaction (Rotor-Gene Q Series, Qiagen) to determine the ninein messenger RNA level in knockdown and non-target cells. Ninein was detected using LightCycler 480 SYBR Green I Master (Roche) in a 20 μl reaction containing primers specific for ninein (dog F ninein: 5′-AGGCCGAGG AGAGCTTTAAC-3′; dog R ninein: 5′-CGCAACTCCTCTTTTTCCAG-3′). Input was determined using dog glyceraldehyde 3-phosphate dehydrogenase (GAPDH) primers as reference (dog F GAPDH: 5′-CATCACTGCCACCCAGAAG-3′; dog R GAPDH: 5′-CAGTGAGCTTCCCGTTCAG-3′). Appropriate no-RT and non-template controls were included in each PCR reaction and dissociation analysis was performed at the end of each run to confirm the specificity of the reaction. The ninein mRNA/GAPDH mRNA content of each sample was determined on the basis of the threshold cycle (Ct) by reading fluorescence of the SYBR Green (reporter). Ct was determined manually where the slope of the fluorescence curve is the steepest and the reaction is linear. Ct for ninein was around 25 cycles indicating low amount of ninein mRNA in si non-target cells. The relative mRNA content was then calculated (ΔΔCt-method) using the equation (1):

$$relRNA = 2^{(-Ct_{Sample,Ninein} - Ct_{Sample,GAPDHInput} - (Ct_{non-target,Ninein} - Ct_{non-target,GAPDHInput}))} \quad (1)$$

where $Ct_{Sample;Ninein}$ is the reverse transcription PCR reaction result of a specific sample using the Ninein primers and $Ct_{Sample;GAPDHInput}$ of the same sample using the dog GAPDH primers for input control. Three independent knockdown experiments each in triplicate were performed.

**MDCK movie analysis.** At the end of imaging, (i) the quality of the raw images, especially at the last imaged time point, and (ii) cell number were evaluated for promising tracking attempts and information gain. This resulted in the tracking of cells only in sufficiently good movies were at least eight cells were present at imaging end.

Raw images of qualifying movies were stacked for each position using a custom-made Matlab code[20]. Each position stack was read into Imaris 7.1.1 (Bitplane, Schlieren, Switzerland) and the nuclei automatically segmented (Surface generation) with local contrast background subtraction and individually adjusted threshold. Tracking of the nuclei was performed automatically (auto-regression motion; maximal distance: 10 μm; maximal gap size: 0). The automated segmentation and tracking results were corrected and verified by three independent researchers. For the manual corrections, the experimental condition (si ninein, si non-target) was blinded for the evaluating person.

Cell IDs and links were exported using a custom made Imaris Xtension script (see ref. [20]). All following calculations were performed in Matlab. In brief, cell lineage trees were generated with nodes representing cell division events and edges representing the cells' lifetime. The lifetimes of individual cells were assessed once. One cell cycle duration is defined as the imaged lifetime of a cell, whose nucleus became just detectable as a discrete fluorescent unit at the end of a mitosis until the time point this cell completes its own mitosis. Mitotic stages were deduced by the characteristic spatial and temporal sequence of histone-labelled chromosomes, whereby neither nuclear envelope reformation nor cell abscission events could be tracked. Cell cycle durations were bijectively assigned to the cell IDs (Fig. 1a).

**MDCK correlations.** Pairs of mother–daughter, daughter–daughter and granddaughter–granddaughter cell cycle durations were compiled from all available data. Out of these arrays, 60 randomly chosen pairs were used to calculate the Spearman's correlation coefficient for the cell cycle durations. This procedure was repeated 100 times and the resulting Spearman's correlation values plotted as boxplots (Fig. 1c)[21].

**Outlier detection method.** The Matlab script for this method is part of the Propagation Analysis Code version 1.0 available at https://github.com/celldiversitylab/Propagation_Analysis_Code. The motif (Fig. 2a), to which a granddaughter set of cell cycle durations at a given threshold belongs to, was determined in a four-step procedure (illustration in Supplementary Figure 5). Shortly, each granddaughter set of cell cycle durations was ordered with increasing durations from cell1 to cell4, containing unique cell IDs. Next, the pairwise MDs, variance scaled distances, were determined for all cell pairs of these four cell cycle durations and the hierarchical clustering started with the lowest MD using single linkage. Next, the kind of motif was assessed as follows: with a defined threshold, expressed as dissimilarity index number (ratio between the highest distance and the second highest distance), a specific motif was assigned according the following rules: if the dissimilarity ratio between the maximal and the second hierarchy is bigger than the chosen threshold value and (i) involved cell4, which is the cell with the highest cell cycle duration, then the granddaughter set belongs to a 3:L motif; (ii) involved cell1, which is the cell with the shortest cell cycle duration in the granddaughter set, then the granddaughter set belong to a 3:S motif; (iii) involved cell1 and cell2 vs. cell3 and cell4 or cell1, and cell3 vs. cell2 and cell4, then a 2:2 motif is present. If none of this applies, the granddaughter set is assigned to the flat motif. For all figures with the exception of Fig. 2c, a threshold of 2 was chosen, as it identifies two sub-clusters with a dissimilarity that is equal or bigger than the sum of the smallest pairwise MDs of the rest of the data.

Studying the link between information propagation and diversity of cell cycle duration, we used a local in-generation reference (among four related

granddaughter cells) to classify outlier cells allowing to robustly define and especially to compare outlier cells in diverse model systems despite (1) the variable dependencies of cell cycle duration on generations (Supplementary Figure 16b–f), (2) their different moments in cell cycle duration distributions and (3) possible differential time dependent changes in cells of each system. Such a study is not possible employing a global approach (based on an entire population distribution, i.e., cell cycle durations across several generations) for outlier classification.

**Propagation test.** The Matlab script for this test is part of the Propagation Analysis Code version 1.0 (available at https://github.com/celldiversitylab/Propagation_Analysis_Code; see Supplementary Figure 7 and its legend). Briefly, permutations: empirical cell cycle durations were 10,000 times permuted without replacement over all lineages within the same generation. The lineage topologies were maintained. For each permutation, each cell ID of a granddaughter set was assigned to one of the three classes (S, L, 0) using the ODM and the frequency of the outlier division types determined (Fig. 3b and Supplementary Figure 5). Thus, for each permutation a division type matrix was established as an intermediate step.

Step (i) Determination of the empirical probability for each division type. The cumulative probability distribution functions (CDFs) were estimated by permutations of cell IDs. Then, the empirical probabilities for each division type was read out and are displayed as a number in each field (e.g., Fig. 3c).

Step (ii) Test for the type I error of each division type. Step (i) was performed on each of 10,000 random division type matrices that are based on permuted cell IDs to estimate the probability distribution functions (e.g., Fig. 3d). The empirical probability of each division type is represented as a red bar of arbitrary height in the same histogram. Next, we identified in which division type the empirical probability laid outside of the two-sided 95% confidence interval of the random scenario; significance can be read out from the indicated p-value. Finally, division types that were significant for the empirical data were counted and used in step (iv).

Step (iii) Test for the type I error of all division types. Using the p-values as criterion as in step (ii), for each permutation, division type and permuted probability distribution, the significantly different division types were determined. These frequencies of the 10,000 permutation tests, where each time a permutation matrix was used as reference to test against, were compiled in a frequency plot with indicated cumulative probabilities of how many division types were laying outside of the 95% confidence interval in the random scenario.

Step (iv) Result of propagation test: *Per definitionem* for this study, only if the empirical results on step (ii) and (iii) were both below the 5% significance level, the division type in the matrix is colour-coded (colour definition in Fig. 3).

**MDCK cell dynamics and colony shape measures.** The periphery index represents the frequency of each cell being at the periphery of the cell colony during its lifetime. The periphery of a colony was estimated using the boundary function in Matlab (shrinking factor 0.5) on all nuclear centroids of a colony. If a nucleus was part of that boundary, it was classified as peripheral.

Cell colony circularity was calculated by the same boundary estimation. The slope of the colony circularity over time was determined to demonstrate the changes over the imaging time of the developing colony (robust linear regression).

Cell velocity was calculated using the Euclidian distances of each nuclear centroid between consecutive time points. The median cell velocity represents the median velocity for each cell over all time points.

**Statistics.** All analyses were performed in Matlab.

No statistical methods were used to predetermine sample size. The degree of freedom values (df) for Fig. 1b is 1 and that of each division matrix 6. The power of the propagation test was computed (Fig. 3e; d-panels of Supplementary Figures 9, 11–15). No raw data were excluded from the analyses, if the criterium for MDCK movie analysis (above) was met and with the exception of differentiating haematopoietic stem cell lineages as follows. Out of 101 blindly picked lineages of the complete published data set of the differentiating haematopoietic stem cell lineages, 30 of those lineages containing cells with the 25 shortest cell cycle durations in the 101 lineage data set were censored, as these probably represent tracking errors. The resulting 90 differentiating haematopoietic stem cell lineages are the basis for the here presented results (Supplementary Table 1). Individual statistical tests performed are mentioned in the figure legends; in general, non-parametric tests were performed.

To test for differences in medians, the Kruskal–Wallis test was applied; for differences in variance, the Levene's test was applied. Multiple comparisons were adjusted for according to the Tukey's honest significant difference criterion if not mentioned differently. Fisher's exact test was applied to test for differences in categorical data. Robust linear regression using a bisquare weight function was applied to test for trends over time or generations. The procedure for testing whether the frequency of empirical division types are different from random is described in the section "Propagation test details".

**Effect size**. Effect sizes were expressed in two types of rz-scores. The rz-score reported in Fig. 1b was calculated according to equation (2):

$$\text{rz-score} = \text{abs}(\text{median}(\text{si non-target}) - \text{median}(\text{si ninein}))/ \\ \text{avr}(\text{MAD}(\text{si non-target}), \text{MAD}(\text{si ninein})), \tag{2}$$

with abs absolute value, avr average and MAD median absolute deviation. "Si non-target" and "si ninein" here refer to the distribution of cell cycle durations of the mentioned condition. The pooled cell cycle durations of the two conditions differed by about 1 MAD. Thus, the effect size between these two conditions is in the same magnitude as half of all data points are distant from a median, which we interpreted as a clear effect.

The individual estimated rz-scores for the division types in the different model systems and conditions displayed in Table 2, reporting the absolute difference between the expected and the observed division type counts divided by the MAD of the expected count distribution, were calculated according to equation (3):

$$\text{estimated rz-score} = \text{abs}(\text{observed} - \text{median}(\text{expected count distribution}))/ \\ \text{MAD}(\text{expected count distribution}), \tag{3}$$

with the expected count distribution corresponding to the corresponding CDF plots of the analysed division type (Fig. 3d and Supplementary Figures 9c, 11c, 12c, 13c, 14c). In general, in all analysed systems and conditions, the division types that were tested significant in the propagation test had the maximal estimated z-scores of the given division type matrix and these values ranged between 3 and 14.5. Thus, we interpreted that our significant propagation tests go along with clear effects.

**Lineage trees of non-si-treated cells**. The lineages of MDCK cells, grown in three-dimensional extracellular matrix, *E. coli*, mouse embryonic stem cell or mouse haematopoietic stem cells published in refs. [20,28–30], respectively, were used to assign cell IDs as in Fig. 1a and their corresponding cell cycle durations. Cell cycle durations of cells where only complete cell lifetimes were imaged were analysed, i.e., period between the birth of a cell at the end of a first mitosis until the cell enters and completes itself mitosis (Supplementary Table 1).

Hermaphrodite embryos of *C. elegans* wild-type strain N2 Bristol were recorded at 25 °C and tracked as described in ref. [34] (Supplementary Table 1).

**Reporting summary**. Further information on experimental design is available in the Nature Research Reporting Summary linked to this article.

## Data availability

The source data underlying Figs. 1–5 and Supplementary Figures 1, 3–4, 6, 8–17 are provided as Supplementary Data 1–8. Other data that support the findings of this study are available upon reasonable request for si non-target and si ninein MDCK tracked lineages from R. Kroschewski (ruth.kroschewski@bc.biol.ethz.ch), for *E. coli* lineages[28] from Joan Slonczewski (slonczewski@kenyon.edu), for *C. elegans* lineages from R. Schnabel (r.schnabel@tu-bs.de) and for mESC[29] and mHSC lineages[30] from T. Schroeder (timm.schroeder@bsse.ethz.ch).

## Code availability

The Propagation Analysis Code version 1.0 for determining the motif of the granddaughter sets and thus outlier cells (ODM), as well as performing the permutations and the propagation test is available at https://github.com/celldiversitylab/Propagation_Analysis_Code.

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

## Acknowledgements

We are grateful for the fruitful comments of Sophie Martin, Mark Robinson, Gabor Szekely and Yves Barral during our analysis. We acknowledge the support of ScopeM/ Swiss Federal Institute of Technology ETHZ. R.K. was supported in part by National Centre of Competence in Research (NCCR) "CO-ME Computer Aided and Image Guided Medical Interventions", the Swiss National Science Foundation (31003A-108160) and the ETH Zurich Research Grant ETH-32 08-3. T. Stadler was supported in part by the European Research Council under the 7th Framework Program of the European Commission (PhyPD; grant agreement 335529). T. Schroeder was supported in part by the Swiss National Science Foundation (156431).

## Author contributions

U.B.: conception and design, acquisition of data, analysis and interpretation of data, drafting and revising the article. D.B.: acquisition of data. T. Stadler: analysis and

interpretation of data, revising the article. A.W. and T. Schroeder: formatted and contributed essential published data. T. Schroeder: revising the article. R.S.: contributed unpublished essential data. R.K.: conception and design, analysis and interpretation of data, drafting and revising the article.

## Additional information

**Competing interests:** The authors declare no competing interests.

