## [Peer Review File · Nature Communications]

Reviewers' comments:

Reviewer #1 (Remarks to the Author):

Berge et al. analyse pedigrees of MDCK cells in terms of cell cycle length. They find that cell cycle length correlations decrease with time and that correlations within a generation (of sister and cousin cells) are larger than across generations (mother to daughter cells). Also, they find outlier cells with much longer cell cycle lengths than other cells. The authors claim that these outlier cells appear more often than expected together with 3 cousins with normal cell cycle lengths, and that this motif appear more often than expected from a mother cell with normal length. Upon siRNA mediated down-regulation of the protein ninein, these effect disappear. The author find long cell cycles in individual cousins during *C. elegans* development and claim that their findings hint towards a cell-autonomous effect mediated by the parent cell.

Analysing single cell pedigrees is a challenging but worthwhile endeavor since this data type becomes increasingly available, making inference of single cell behaviour and memory possible. While the paper addresses an interesting and potentially important effect that might explain cellular mechanisms required during high proliferation, it fails to convince me that the effect is as ubiquitous as claimed, and has substantial shortcomings in the presentation and analysis of the data.

From my perspective a more concise theoretical framework, in particular more sophisticated null models, and more data from the studied cell system and other systems is required to substantiate the conclusions of the authors.

*** Major issues

**** Methodology

- Much of the later analysis of 'outlier cells' is based on the observation that some cells have a much longer cell cycle length than others. While this is not obvious in Fig. 1A, Fig. 2B suggests a bimodal distribution of pairwise cousin correlation coefficients. I would like to see if this bi-modality is visible in 3 independent experiments.
- In the tree data, it is difficult to define a proper null model to compare the observations with what one would expect. The assumption that 'cell cycle durations are uniform' is extremely naive though. What if one draws from the observed distribution of cell cycle times? Maybe specifically for the generation the cousin cells observed are in. Would the number of 3L cousin groups be still higher than expected?
- Tracking was done automatically via the Imaris software. Although the authors claim that 'tracking results were corrected and verified by three independent researchers' (what if they did not agree?), I am concerned about the data quality. Are the authors sure about cells that divide within 3.5h? A supplementary movie would help here and would also prove tracking accuracy.
- The authors claim that 'qualitative inspection' of other data sets substantiates their claim. I think a quantitative analysis of more

experiments (replicates of the system studied, but importantly also without siRNA transfection, and other data sets) is needed for that.

**** Presentation

- Fig. 3 is mostly explaining the method to generate the permutation test. This should go to the supplements I suggest. The interesting part 3B lacks a legend, but also examples from the data that illustrate the under and over represented transitions.
- I got lost during reading the manuscript when it came to 3L cells, 3:L motifs and O/L division types. A more concise nomenclature, and maybe a focus on one strong effect could help.

*** Minor issues

**** Presentation

- What are the '36 movies' on page 5? 36 pedigrees?
- Fig 1: 'G1' in A is misleading and not consistent. In D, 'generation' is used.
- Fig 1D: What does 'n lineages' mean? The number of cells?
- The authors used widefield imaging? The reference to light sheet paper of Pampaloni et al. is misleading then.
- Please show scatter plots of correlations between the pairs shown in Fig. 2A.
- Fig. 2E: 'lineage ID23' has no meaning for the reader.
- Fig S4: the last row in A is redundant.
- (S/S) transitions on page 8 are not analysed in the figure.

**** Text

- The nomenclature used is inconsistent: What are the 'control lineages' (page 6)? What exactly is defined as a 'motif'? The authors use 'cell cycle motives', 'diversity motives', 'special motives' etc.
- Do all animal cells divide asymmetrically, as claimed in the intro?
- On page 4, the authors say that cell 'initially' form small colonies. What happens later? Pls provide movies showing cell growth over the complete experiment time.

**** Methodology

- Fig 1D: Why is the data subsampled 'if the total number of data points was bigger than 14'? IQRs can be calculated directly from the data I guess.
- The grey background of 0.4 in 2B is arbitrarily chosen, right?
- Instead of choosing 100 cell pairs at random, standard bootstrapping could be applied to estimate error bars in Fig. 2A
- Fig 2A: 'Median p-value' sounds odd. There should be a p value describing if the correlation is sig. different from 0 I guess.
- Fig. 2D: Are these multiple Fisher tests?

Reviewer #2 (Remarks to the Author):

The manuscript presented by Berge et al. is an interesting piece of work aiming at identifying mechanisms that may account for cell diversity traits such as long-known cell-to-cell heterogeneity

observed in many systems. This paper addresses in particular the question whether cell cycle length variations within a population of symmetrically (in terms of cell fate) dividing cell culture cells occur stochastically or if they are rather the consequence of a non-random process such as an asymmetric division of a parental cell. To this end the study uses long-term imaging of single MDCK cells tracing the lineage and monitoring cell-cycle length of each cell while small colonies form. This data set documenting the division history of cells within a lineage over up to 7 generations was then subjected to thorough statistical analysis asking the question whether cell cycle length variations occur at random or whether specific patterns can be identified that correlate with the occurrence of cell cycle length differences. The key finding is that between four cells that are related as "granddaughter cells" a pattern can be identified, where three cells have rather the same cell cycle length, while there is one outlier with typically a much larger cell cycle (called "3:1 motif", the outlier "L-cell"). This pattern is lost in lineages that are depleted for Ninein, suggesting that normally a Ninein-dependent process is required such that a parental cell divides to produce daughter cells that are asymmetric in terms of cell cycle length.

This is a very interesting and timely topic and the study is carried out at a good standard with generally valid experiments and conclusions. Criticism could be that the evidence for the 3:1 motif is shown for two examples, relativizing the general validity of the claim. I also find the manuscript hard going and quite technical in places, the quite extensive use of statistic jargon is not helpful. Nonetheless, I would support an invitation for revision.

It would be great to substantially improve the flow of the arguments. For instance, in the introduction a clearer separation of cell fate and cell diversity/heterogeneity should be provided. I agree that both may be linked, but for the sake of the paper, dealing with heterogeneity, no evidence for cell fate relevance expect for correlations, this would be helpful. Also, the relevance of different age of centrioles needs to be better introduced better, it comes out of nowhere and "Accordingly" is rather confusing here. I am also not sure how robustly it is established that all cells divide asymmetrically in terms of centriole age segregation.

While Ninein certainly has an effect, I am not too sure that this is a robust test for a cell autonomous effect and ultimate proof of an asymmetric division. MDCK cells in the presence of serum do have a primary cilium if I am not mistaken, which depends on Ninein. How can it be ruled out that Ninein depletion disrupts a cilium dependent process in interphase that affects the cell cycle length?

The 3:1 is a robust pattern that can be identified and based on statistical testing does not occur at random frequencies, yet - unless I missed something - at least in MDCK cells its occurrence cannot be predicted (e.g. Fig 3A1, L-cells have the same empirically determined probability to divide to give rise to O/O or L/O cells). Therefore, in principle the very first cell imaged could be a cell that produces two daughter cells with a very different cell cycle (the granddaughter pair with an outlier). Yet Fig 2A suggests that cell cycle length variation in daughter-daughter pairs (as which the one I am referring to would belong) is low. What is the variation of cell cycle length in daughter cell pairs of the first generation only? Does this matter at all, or does the cell culture somehow resets this behaviour? Is there a culture effect also in cell cycle decrease? What is the explanation for that, this is rather counter-intuitive.

The *C. elegans* data is interesting, but remains a correlation. We know that these lineages are non-random (a series of well characterized asymmetric divisions leads to the generation of the germ line). One can even find the longer cell cycle lengths of the cells in the P lineage in worm book. So, is it surprising that this lineage follows a pattern as found here? These divisions are also asymmetric in size (The P0 cell is smaller than the AB cell for instance, and there is a clear effect on cell size and the length of the cell cycle (e.g. Arata et al Front. Physiol. 2015). So, cell cycle length is perhaps a consequence of cell size and more factors than cell cycle length are involved in specifying this lineage and its potential. In the P lineage, it appears that cell cycle length can serve as a predictor for another division producing cell cycle length variations, which is also different to

MDCK cells. Perhaps a bit more cautious discussion is needed here.

minor

Summary:

"its" third line– not clear what this refers to

".but appeared to emerge from asymmetric divisions taking place at non-random levels: a parent cell determines the unequal partitioning of the cell cycle duration in its two progenies". Can you really say that, as there is no way to predict which parental cell will do it? What is "levels" referring to, this would be important to elaborate in the discussion.

I think the figure legends could generally use some work to clarify, what is shown.

Fig 1C:

This excludes cell1 and cell 1112? Correct? Please clarify in legend.

Fig S7:

"Representative immunofluorescence images of si non-target and si ninein cells. The cells labelled for ninein (green), γ -tubulin (red) and nuclei (blue) show a lack of centrosomal staining in depleted cells at 0 h (94% +/- 0.88%) and 24 h (80.7% +/- 1.76%) but not at 100 h (5.1% +/- 3.18%) after imaging started."

- C shows 24h,48h,124h not 0h, 24h, 100h, so what is it?

If the duration of the si ninein effect wears dramatically off at ~100/124h, and the imaging goes for 85h it may be good to include 90h of si ninein as a control, showing that the depletion is still efficient.

Reviewer #3 (Remarks to the Author):

It is known that in overtly asymmetrically dividing cells, a variation in cell cycle duration is seen between the two daughters. This paper asks whether a similar asymmetry between daughters is established at the time of division even in lineage where cell division is apparently symmetrical. In this way the authors seek to identify a cryptic asymmetry through careful mathematical analysis of the temporal aspect of lineages. This is a very clever and interesting idea. I really like this paper, I think it is a valuable addition to the cell heterogeneity literature and will attract a broad readership. Its elegance reminds me of the work by the "phage group" of Delbrueck et al back in the early days of molecular biology.

The main approach is a computational search for cell division motifs in lineage data. The description of this analysis is a bit complicated, because the analysis IS complicated, but I feel that the authors did an excellent job in motivating the idea of what they are doing so that readers who don't want to fully delve into the details are able to still understand the general idea, and for those who do want to delve into the details, I felt that the procedure was carefully described such that it would indeed be possible to replicate it based solely on this description.

The inclusion of ninein RNAi as a control is extremely important for several reasons. First of all it gives insight into potential cell biological bases for the asymmetry, but I think even more importantly, with what is essentially a pattern detection algorithm, the question is always how likely would such patterns be accidentally detected when in fact there is nothing there to detect, i.e. what is the false positive rate for detecting a certain type of motif. There are various ways to

try to generate synthetic data or otherwise estimate such errors, but it is always far more convincing to be able to provide a negative control data set, and then analyze that for comparison.

The authors did a good job of exploring several possible bases for the 3L motif, ruling out variation in mitotic timing or position in the clone. However, they do not appear to have investigated cell size, or at least I could not find any place where this was discussed. The model that I have in mind is that perhaps a cell is born that is unusually small. The mechanisms that regulate cell size are, of course, not really known, but in general there is the idea that cells may grow to a defined size before they are able to divide. Such a "sizer" model for cell size regulation would predict that cells born small would have a longer interphase before their next division. By the same token, if the 3,L pattern is not accompanied by variation in at-birth cell size, it would be interesting to know if the cells which grow longer end up larger, because they have had more time to accumulate biomass. I feel that even a simple quantification of at-birth and at-division cell size might be of great interest to the cell size community and could reveal new insights into this question, possibly helping to resolve the ongoing debates about sizers versus timers versus adders. This is just a suggestion, I wouldn't want to force the authors to do this, but I think it might be extremely interesting.

I am confused by the transition rate analysis of figure 3. I would have thought that the rate of O/L should be balanced by the rate of L/O, since asymmetric transitions would suggest that over time more and more 3,L motifs should be occurring. But this could well be a confusion on my part as to how the transition matrix is constructed. I am interpreting the matrix as a standard Markov transition matrix, which would be multiplied by the distribution of motifs at a given generation to find the distribution at the next generation. I think perhaps the matrix given here is actually the product of such a Markov transition matrix with the distribution vector, i.e. the matrix entries are weighted by the probability of a given initial motif. I am not familiar with the term "empirical transition matrix" but I am getting the impression that this is what that term describes. I would therefore suggest that the authors might want to be a little bit more explicit about what these matrix entries mean. i.e., they do not mean the probability that a cell from a given motif will produce progeny showing another motif, but rather the probability that a cell chosen from a population is part of a certain motif AND that it then produces progeny that are part of a certain motif. So that would explain why there are so many more O/L than L/O, simply because there are more O than L in the population. But then I do wonder, if the authors were to correct for the abundance of the initial motifs, do they now see an equal conditional probability that a cell from a O motif produces L daughters as that a cell from an L motif produces O motif daughters? I Otherwise, I would be curious to know whether there is a different overall abundance of L motifs in older colonies.

The results with *C. elegans* are potentially complicated by the fact that each cell in the lineage is known to be undergoing a different fate decision than the others, i.e. it is not a homogenous background. So personally I find the MDCK results much more interesting. But there is nothing wrong with the *C. elegans* data.

Very small comment:

1. The patterns are described as "motives" in some places, and "motifs" in others. I believe that the word "motive" should be replaced with "motif".

Detailed responses to the referees:

Answers to the comments made by Referee 1.

Berge et al. analyse pedigrees of MDCK cells in terms of cell cycle length. They find that cell cycle length correlations decrease with time and that correlations within a generation (of sister and cousin cells) are larger than across generations (mother to daughter cells). Also, they find outlier cells with much longer cell cycle lengths than other cells.

The authors claim that these outlier cells appear more often than expected together with 3 cousins with normal cell cycle lengths, and that this motif appear more often than expected from a mother cell with normal length.

Upon siRNA mediated down-regulation of the protein ninein, these effect disappear.

*The author find long cell cycles in individual cousins during *C. elegans* development and claim that their findings hint towards a cell-autonomous effect mediated by the parent cell.*

Analysing single cell pedigrees is a challenging but worthwhile endeavor since this data type becomes increasingly available, making inference of single cell behaviour and memory possible. While the paper addresses an interesting and potentially important effect that might explain cellular mechanisms required during high proliferation, it fails to convince me that the effect is as ubiquitous as claimed, and has substantial shortcomings in the presentation and analysis of the data.

From my perspective a more concise theoretical framework, in particular more sophisticated null models, and more data from the studied cell system and other systems is required to substantiate the conclusions of the authors.

Reply: We would like to thank the referee for taking the time to reflect on our study and to generally support the analysis of single cell lineage trees. We further thank the reviewer for highlighting some important shortcomings of the manuscript, as addressing them enhanced the impact of our initial finding e.g. other cell systems. According to her/his concerns, we are presenting lineage analyses of three additional cell systems (also Referee 2 had asked for this). The prokaryote *E. coli*, is one of the additional systems. As it is evolutionary very distant from MDCK it strongly supports the conservation of the 3:1 diversity motif and thus its biological significance. We also added more data of the MDCK cell system (Figure S13a) and altered the null hypothesis to study the frequencies of the motifs (Figure 1c,d in the current manuscript). Yet, we wish to clarify that this is not a theoretical study but one that tests the working hypothesis that division asymmetry could be a cause for observed heterogeneity of cell cycle durations, we combined statistical with biological arguments.

*** Major issues

**** Methodology

- Much of the later analysis of 'outlier cells' is based on the observation that some cells have a much longer cell cycle length than others. While this is not obvious in Fig. 1A, Fig. 2B suggests a bimodal distribution of pairwise cousin correlation coefficients. I would like to see if this bi-modality is visible in 3 independent experiments.

Reply: To address the referee's concerns we illustrated our point better already in Figure 1 of the current manuscript. We present now the first lineage with green marked long outlier cells (Figure 1a, in the current manuscript (Figure 1A of previous submission)). We also wish to refer to Figure S3e in the current manuscript (Figure 2E of previous submission), where an additional lineage is depicted with four long outlier cells (labelled "L").

Regarding the request of per experiment analysis of the histogram analysis of Figure 2b of the original manuscript (Figure SF3b in the current manuscript). The basis for the si non-target data set are 4 independent experiments and 5 independent experiments for the si ninein data set as documented in Figure S2 in the current manuscript (Figure S2 of previous submission). Further, we wish to clarify, that we had pooled data of all experiments, bootstrapped from this pool and analysed all pairwise daughter-daughter and granddaughter cells pairs of a given condition. We had chosen this approach to balance for our strongly varying empirical input numbers (those present as daughter-daughter pairs in the lineages) within our experiments and across conditions (SF3a in the current manuscript and Referee Figure 1 here in the appendix) as Spearman's correlation coefficient values are dependent on the input number. Therefore, a comparison of Spearman's correlations coefficients by disentangling the experiments would not be possible.

To still respond to the referees point we chose a different way to display the results of an equivalent analysis. We analysed per experiment the diversity of cell cycle durations amongst the empirical daughter-daughter pairs in the si non-target and si ninein data sets. We focused on all daughter-daughter pairs in the data as the pairs amongst granddaughter cells is included in such an analysis and the data volume bigger and thus more representative.

Please find here in the appendix in Referee Figure 1 the results of such an analysis. In all conditions and experiments daughter-daughter pairs are found that correlate well (i.e. are close to the line with slope 1), while others do not (above the orange line indicates at least 25% difference in cell cycle durations between the two daughter cells). Consistent with our pooled and bootstrapped histogram data focusing on pairs of granddaughter cells (SF3b in the current manuscript) in most experiments the divergent cell pairs were stronger divergent in the si non-target condition (further away from the orange line) than in the si ninein condition, where all daughter-daughter pairs were in vicinity to the red and below the orange line. This analysis documents in individual experiments a stronger heterogeneity of cell cycle durations in daughter-daughter pairs in the si non-target condition compared to the si ninein condition.

- In the tree data, it is difficult to define a proper null model to compare the observations with what one would expect. The assumption that 'cell cycle durations are uniform' is extremely naive though. What if one draws from the observed distribution of cell cycle times? Maybe specifically for the generation the cousin cells observed are in. Would the number of 3L cousin groups be still higher than expected?

Reply: We thank the referee for this excellent idea, which we followed: we randomized cell cycle duration within individual generations. The novel data is depicted in Figure 1c,d in the current manuscript. We find that the 3:L motif in the si non-target condition is up to a dissimilarity index of 3.5 above the estimated confidence interval. However, for si ninein this motif is for all the tested dissimilarity indices never outside the 95% confidence interval of a random scenario. Thus, indeed, the number of 3:L motifs is greater than expected in si non-target MDCK cells also when we look at cell cycle length distributions within a given generation.

In the original manuscript, we had already used the exact same randomization mode for the propagation analysis, a central test of this study. There (now Figure 2; Figure 3 of the original manuscript) the cell cycle durations of all lineage trees were permuted only within individual generations and across the lineages. Importantly in Figure 2 and related figures (Figures 3a, 4, S7-12) the null model tests the biologically critical question if cell cycle duration variability is propagated across generations. This study aims to use statistics to analyse if cellular divisions have somehow an impact on the variability of cell cycle durations in a near wildtype condition and to compare the obtained results to those obtained after experimental manipulation (si ninein). Hence, the statistics results are biologically controlled, which eliminates the need for more theoretical controls.

- Tracking was done automatically via the Imaris software. Although the authors claim that 'tracking results were corrected and verified by three independent researches' (what if they did not agree?), I am concerned about the data quality. Are the authors sure about cells that divide within 3.5h? A supplementary movie would help here and would also prove tracking accuracy.

Reply: We understand the concerns of the referee. Tracking the cells is very difficult especially if the colony increases in size. Firstly, we went back to the movie with the 3.5 h cell cycle duration and display here in the appendix in Referee Figure 2 at the end of this document the critical sequence of still images and the way we assigned the critical cell. In the method section of the revised manuscript we clarify with the following word how we define cell cycle duration, namely: "One cell cycle duration is defined as the imaged lifetime of a cell, whose nucleus became just detectable as a discrete fluorescent unit at the end of a mitosis until the time point this cell completes its own mitosis. Mitotic stages were deduced by the characteristic spatial and temporal sequence of histone-labelled chromosomes whereby neither nuclear envelope reformation nor cell abscission events could be tracked."

To assess the impact of short cell cycle durations on our propagation analysis we provide the corresponding results in the appendix for the referee (see Referee Figure 3 below). Omitting the 5 or 10 shortest cell cycle durations per condition does not abolish the determinism in the propagation test for si non-target and also does not alter the result for the si ninein condition. The alterations in the si non-target condition compared to the data set presented in the manuscript (Figure 2 in the current manuscript) are due to a reduction of divisions that could still be analysed (from 70 (Table1 supplementary information, Figure 2 in the current manuscript) to 64 after removal of the 5 shortest and 62 after removal of the 10 shortest cell cycle durations). Thus, these possible tracking errors have little effect on our propagation analysis and none on the interpretation of the results.

Most importantly, any inaccuracy in our tracking should be in the same range in both si non-target and si ninein conditions. Thus, the conclusions derived from our comparative statistics and complemented with biological environmental tests still hold.

In the current manuscript we now provide two exemplary movies as supplementary material for the reader to make his/her own judgement about the quality of our imaging and tracking method (supplementary movie 1, a si non-target cell divides over time; supplementary movie 2, a si ninein cell divides over time). In both movies we show side by side the fluorescence channel (left) and the tracking marks superimposed on the fluorescence channel (right).

We want to point out that such short cell cycle durations have already been reported in the scientific literature and seem to indeed report true events. For example, the cell cycle duration of *Drosophila* neuroblasts is less than 2 hours ¹, and that of embryonic mouse epiblast cells is estimated to be comprised between 4.5 and 8 hours ². Interestingly, murine neuronal precursor cells increase the duration of interphase from 11.1 h to 17.4 h (average values) as they progress towards neuronal differentiation ^{3,4}.

In addition, we have now added and analysed data obtained from others with a different imaging set up and tracking methods, which are all like ours at the end manual methods. The conclusion obtained from these new datasets support the conclusions obtained with our system. Thus, the presence of the 3:L motif is not a peculiarity coming from our experimental workflow (see Table 1 in supplementary

material in in the current manuscript). Specifically, we now include the analysis of 7 embryonic lineages of *C. elegans* imaged and tracked as described in ⁵. We also include lineage data from *E. coli* (2 conditions, total 17 lineages; Figures SF 8,9,13a,c in the current manuscript, ⁶), and mouse embryonic stem cells (5 lineages, Figures SF10, 13a,d in the current manuscript, ⁷), mouse haematopoietic stem cells (90 lineages, Figures SF 11, 13a,e, in the current manuscript, ⁸). Thus, although we cannot exclude tracking errors, our 3:1 motif is present in these four non-MDCK model systems and most importantly propagation of variability of cell cycle durations can be modulated in *E. coli* like in MDCK.

- *The authors claim that 'qualitative inspection' of other data sets substantiates their claim. I think a quantitative analysis of more experiments (replicates of the system studied, but importantly also without siRNA transfection, and other data sets) is needed for that.*

Reply: This point is very well taken. As mentioned above, we now include a quantitative analysis of many additional trees taken from various systems (see above and additionally MDCK cells without siRNA transfection (Figure SF13a in the current manuscript), overview of all analysed systems in Table1 supplementary information in the current manuscript). However, we wish to clarify, that data of 4 independent si non-target MDCK and data of 5 independent si ninein MDCK experiments were already in the previous submission included (detailed now and in the previous submission in Figure SF2a,b). This fact is more highlighted in the current result part of the manuscript (section "A 3:L cell cycle duration motif is frequently present in lineages of single MDCK cells").

**** *Presentation*

- *Fig. 3 is mostly explaining the method to generate the permutation test. This should go to the supplements I suggest. The interesting part 3B lacks a legend, but also examples from the data that illustrate the under and over represented transitions.*

Reply: Since *Nature Communication* targets a wide readership, supports transparency of methodology, and biologists and bioinformaticians or systems biologists might well appreciate the display of the for them unusual analysis, we opted against the suggestion of this referee and maintained the figure part in place. Yet, to accommodate the referee's recommendation we added an illustrative example (Figure 2a, in the current manuscript). Other division type examples can be seen in Figure S3a, in the current manuscript (Figure 2e in previous submission). Specifically, there are examples for four 0 to 0/0, two 0 to L/0 and one L to 0/0 divisions. This figure is a novel statistical analysis for the cell cycle field, central to our results and thus important to present. Especially, it shows in detail how the empirical values of the frequency of si non-target MDCK division types relate to randomization results in the propagation analysis (Figure 2d, in the current manuscript). Additionally, the legend for Figure 2f in the current manuscript reads now as follows "Illustration of the propagation test result: The empirical values for the division types 0 to S/0 and 0 to L/0 are significantly unlikely. Only if the 5% probability criteria in d) and e) were met, such a matrix is colour-coded: red - most likely overrepresented, green - most likely underrepresented division types." and should be understandable as it follows the other legend parts.

- *I got lost during reading the manuscript when it came to 3L cells, 3:L motifs and 0/L division types. A more concise nomenclature, and maybe a focus on one strong effect could help.*

Reply: We can empathize with the referee and simplified the presentation in the revised version of the manuscript. To facilitate the reading we i) added illustrations as in Figure 1c and 2a, in the current manuscript, ii) simplified the text moving rare and complex cases with an illustration to the supplementary Figure S6b in the current manuscript.

*** *Minor issues*

**** *Presentation*

- *What are the '36 movies' on page 5? 36 pedigrees?*

Reply: The full sentence reads “From the 36 movies acquired of cells with the si non-target treatment in 4 independent experiments, we generated lineages and analysed the cell cycle durations (Figures 1a, S2, see Methods and ⁹). In these 36 lineages ...” We use intentionally the term “movies” to convey the notion that time resolved images of the dividing cells, thus movies, are at the beginning of the analysis and thus of a lineage tree.

- *Fig 1: 'G1' in A is misleading and not consistent. In D, 'generation' is used.*

Reply: We agree. An accordingly altered version of that figure part is now presented in Figure 2a in the current manuscript.

- *Fig 1D: What does 'n lineages' mean? The number of cells?*

Reply: Thank you, this question had us realize our unusual display, which we now altered in the figure (now Figure 1b), figure legend and figures with a similar display. The revised legend part for Figure 1b reads now “ Below: lineages - number of analysed lineage trees; data points (blue) - number of cell cycle durations; median - median cell cycle duration; IDR/median - interdecile range normalised to the median as quantifier for dispersion.” The number of analysed cells is not depicted but the number of cell cycle durations, which are defined as completely imaged lifetimes of cells.

- *The authors used widefield imaging? The reference to light sheet paper of Pampaloni et al. is misleading then.*

Reply: Yes, indeed we used widefield imaging to follow the si non-target and si ninein MDCK cells. We refer to the light sheet paper of Pampaloni et al. not for the imaging setup but for the data extraction from images, which was introduced at this step in the manuscript.

- *Please show scatter plots of correlations between the pairs shown in Fig. 2A.*

Reply: Thank you, for this very good suggestion. We altered the figure part accordingly (now Figure S3a in the current manuscript). As a result, the bimodality amongst granddaughter-granddaughter pairs became now also visible in this part of the figure (Figure S3b in the current manuscript). Though in the histogram version (Figure S3c) it is easier detectable.

- *Fig. 2E: 'lineage ID23' has no meaning for the reader.*

Reply: For transparency reasons, we kept this information in the legend text (now Figure S3e). Lineage ID 23 is for example also depicted in Figure S2a in the current manuscript and supports the transparency of the analysis.

- *Fig S4: the last row in A is redundant.*

Reply: We now represent the revised data in a different manner (Figure 1c,d in the current manuscript) and leaving out this redundancy.

- (S/S) transitions on page 8 are not analysed in the figure.

Reply: Thank you for pointing this out. It let us realise that there was a mistake in the text, which is now corrected.

Throughout the manuscript: S-cells are by definition part of the 3:S motif, which is a granddaughter set i.e. 4 most closely related cells. Therefore, it is impossible that two daughter cells can be S-cells (see also Figure 2a, for illustration). In other words, that a mother cell divides into two S-cells cannot exist. In the current manuscript we state as follows: "...we obtained the empirical frequency of the nine possible division types, whereby S-, L- and 0-cells divide into (S/0), (L/0) or (0/0) pairs of daughter cells (Figure 2a, b; other division types do not exist due to the nature of the 3:1 motif)"

To clarify further, also in the current manuscript we state explicitly in the figure legend of Figure 2b "Note: a cell dividing into S/S, L/L, or S/L daughters is impossible." (similarly in the legend of Figure 3 of the previous submission).

**** Text

- The nomenclature used is inconsistent: What are the 'control lineages' (page 6)? What exactly is defined as a 'motif'? The authors use 'cell cycle motives', 'diversity motives', 'special motives' etc.

Reply: We thank the referee for pointing out these difficulties.

The term "motif" (plural: motifs) in music describes the shortest melodic or rhythmic figure in a music piece. We use it in this sense now consistently throughout the manuscript.

The term "control lineages" referred to the si non-target lineages, as these are the control to si ninein lineages in the previous manuscript version. In the current manuscript we turn it around and present the si ninein condition as the biological control for the results obtained with the si non-target data set.

Diversity motifs – are motifs that introduce diversity in the cell cycle durations in a group of cells. The most minimalist ones amongst 4 related cells are defined in the current manuscript as follows: "The theoretically most minimalist diversity motifs amongst the cell cycle durations of four cells are the following two: I) a 3:1 motif, where one of the cells has a very different cell cycle duration (outlier cell) relative to the other 3 and II) a 2:2 motif, where the cell cycle durations are more similar within a pair than between the pairs." For illustration we now included this term in Figure 1c in the current manuscript.

"Because cells of the same generation rank are thus more likely to be similar, diversity motifs among these cells are particularly meaningful. Consequently, we looked whether these pairs formed special motifs." Here "special motifs" refers to our search if any sort of motif is present within cells of the same generation and possibly of such a structure that it introduces diversity.

- Do all animal cells divide asymmetrically, as claimed in the intro?

Reply: We thank the referee for this critical question, that let us realise a mistake. Possibly the referee is referring to our centrosome example. As animal cells without centrioles exist, like the oocytes during meiotic divisions, we altered our statement in the current manuscript to "Thus, at least in relation to centriole age most animal cells divide asymmetrically."

- On page 4, the authors say that cell 'initially' form small colonies. What happens later? Pls provide movies showing cell growth over the complete experiment time.

Reply: The text reads now "In both conditions, single cells divided forming colonies during imaging." Additionally, in the current manuscript we provide Supplementary movie 1 and 2 showing in each of them how one initial cell forms initially a small and later a bigger colony.

**** Methodology

- Fig 1D: Why is the data subsampled 'if the total number of data points was bigger than 14'? IQRs can be calculated directly from the data I guess.

Reply: We thank the referee for this suggestion. In the current manuscript we omitted the previous analysis and only present a directly calculated IQR normalized to the median of the cell cycle durations per condition in Figure 1b.

- The grey background of 0.4 in 2B is arbitrarily chosen, right?

Reply: Yes, this border is arbitrarily chosen (Figure S3b in the current manuscript).

- Instead of choosing 100 cell pairs at random, standard bootstrapping could be applied to estimate error bars in Fig. 2A

Reply: We are not aware of «standard» bootstrapping. However, by choosing 100 cell cycle durations of cell pairs at random, analysing these, and repeating these two steps, we indeed bootstrapped to estimate the correlation coefficient distributions. Error bars are a graphical representation of the variability of data. Traditionally this kind of representation is used when indicating SD or SEM. These two measures however, require normality of the data, which is a wrong assumption for most biological data including ours. Therefore, we opted here (now Figure S3a) and throughout the paper for another non-parametric representation of the variability: boxplots, representing the 1st, 2nd and 3rd quartile with whiskers depicting the 1.5 times the interquartile range (as in detail described in the legend of Figure 1b, for all following box plots in the manuscript)

- Fig 2A: 'Median p-value' sounds odd. There should be a p value describing if the correlation is sig. different from 0 I guess.

Reply: For each bootstrap one Spearman's correlation value and one p-value is obtained. Therefore, by bootstrapping 100 times, we obtained 100 p-values. The median p-value is the median of those 100 bootstrapped p-values. However, as this median p-value is not adding important information and is not essential for our argumentation, we removed this label in Figure S3a in the current manuscript in the hope to reduce the complexity of our study.

- Fig. 2D: Are these multiple Fisher tests?

Reply: In the current manuscript this figure part is in Figure S3d. Two Fisher tests were performed: one for si non-target, and one for the si ninein data set comprised of only the cell cycle durations of granddaughter sets. This means we performed for both the si non-target and the si ninein each a Fisher test based on a 3x3 contingency table (see legend of Figure S3d).

Answers to the comments made by Referee 2.

The manuscript presented by Berge et al. is an interesting piece of work aiming at identifying mechanisms that may account for cell diversity traits such as long-known cell-to-cell heterogeneity observed in many systems. This paper addresses in particular the question whether cell cycle length variations within a population of symmetrically (in terms of cell fate) dividing cell culture cells occur stochastically or if they are rather the consequence of a non-random process such as an asymmetric division of a parental cell. To this end the study uses long-term imaging of single MDCK cells tracing the lineage and monitoring cell-cycle length of each cell while small colonies form. This data set documenting the division history of cells within a lineage over up to 7 generations was then subjected to thorough statistical analysis asking the question whether cell cycle length variations occur at random or whether specific patterns can be identified that correlate with the occurrence of cell cycle length differences. The key finding is that between four cells that are related as “granddaughter cells” a pattern can be identified, where three cells have rather the same cell cycle length, while there is one outlier with typically a much larger cell cycle (called “3:1 motif”, the outlier “L-cell”). This pattern is lost in lineages that are depleted for Ninein, suggesting that normally a Ninein-dependent process is required such that a parental cell divides to produce daughter cells that are asymmetric in terms of cell cycle length.

This is a very interesting and timely topic and the study is carried out at a good standard with generally valid experiments and conclusions. Criticism could be that the evidence for the 3:1 motif is shown for two examples, relativizing the general validity of the claim. I also find the manuscript hard going and quite technical in places, the quite extensive use of statistic jargon is not helpful. Nonetheless, I would support an invitation for revision.

It would be great to substantially improve the flow of the arguments. For instance, in the introduction a clearer separation of cell fate and cell diversity/heterogeneity should be provided. I agree that both may be linked, but for the sake of the paper, dealing with heterogeneity, no evidence for cell fate relevance expect for correlations, this would be helpful. Also, the relevance of different age of centrioles needs to be better introduced better, it comes out of nowhere and “Accordingly” is rather confusing here. I am also not sure how robustly it is established that all cells divide asymmetrically in terms of centriole age segregation.

While Ninein certainly has an effect, I am not too sure that this is a robust test for a cell autonomous effect and ultimate proof of an asymmetric division.

Reply: We thank the referee for the very positive and helpful comments. We address the spirit of her/his points by revising the manuscript text and including the analysis of additional cellular systems (see also Referee 1). Specifically:

We altered the introduction in the current manuscript being aware that the term cell fate is so far molecularly hard to mirror. The link between centriole age and cell fate is based on correlations but interesting ones that prompted us to study the situation after ninein downregulation. Indeed as already pointed out in our reply to referee 1, not all cells have centrioles, thus we altered the introduction text accordingly.

We did neither study nor focus on the function of the protein ninein itself in our analysis. The ninein downregulation condition serves in our study as a biological control (as emphasized by Referee 3) to validate and understand the events in the si non-target data set; any other experimental manipulation with similar effects could have served this purpose. In the current manuscript we therefore restructured the line of argumentation to highlight this logic. To remove the focus from ninein further we present now the analysis of *E. coli* lineages. *E. coli* does not express ninein, but a pH-change can modulate the determinism in the propagation analysis (Figure 4a in the current manuscript).

Finally, we agree that we do not present an ultimate proof of an asymmetric division, as we can only point to asymmetric division types with a likelihood but not certainty. In contrast we claim - with only 0.005 error probability (or 99.995 certainty) - that the division types involving special outlier cells are

non-randomly occurring in si non-target lineages. Due to the properties of the outlier motif and absence of detectable external influences asymmetric cell divisions are deduced and now supported by Figure S14 in the current manuscript.

MDCK cells in the presence of serum do have a primary cilium if I am not mistaken, which depends on Ninein. How can it be ruled out that Ninein depletion disrupts a cilium dependent process in interphase that affects the cell cycle length?

Reply: We thank the referee for this very interesting link to the cilium, which we mention in the revised discussion part. However, as already mentioned, we find also significant propagation of cell cycle diversity in *E. coli*. The fact that *E. coli* does not build a cilium but has bacterial flagella points to a potential fruitful comparison in the future. The cellular effects in the si ninein condition are for the conclusions of this study irrelevant as long as it does not alter the gross cellular behaviour like cell cycle progression, cell positions in the growing colony and cell motility. We indeed could not detect such or differential changes between si non-target and si ninein cells.

Ninein downregulation reduced cilium formation or maintenance in mammalian cells as reported by ¹⁰. This effect was observable after 48 h in a serum starvation condition. However, in our cultures 20% FCS and insulin was present, which does not represent a starvation condition.

Our data and publications support the motion that the differences in cell cycle durations arise from altered lengths in the G1 phase when the primary cilium is formed. Thus, the cilium still remains an attractive entry point for future studies in mammalian cells.

The 3:1 is a robust pattern that can be identified and based on statistical testing does not occur at random frequencies, yet - unless I missed something - at least in MCDK cells its occurrence cannot be predicted (e.g. Fig 3A1, L-cells have the same empirically determined probability to divide to give rise to 0/0 or L/0 cells). Therefore, in principle the very first cell imaged could be a cell that produces two daughter cells with a very different cell cycle (the granddaughter pair with an outlier). Yet Fig 2A suggests that cell cycle length variation in daughter-daughter pairs (as which the one I am referring to would belong) is low. What is the variation of cell cycle length in daughter cell pairs of the first generation only? Does this matter at all, or does the cell culture somehow resets this behaviour? Is there a culture effect also in cell cycle decrease? What is the explanation for that, this is rather counter-intuitive.

Reply: We appreciate very much the in-depth thoughts of this referee.

First we wish to clarify:

- a) *"The 3:1 is a robust pattern that can be identified and based on statistical testing does not occur at random frequencies, yet - unless I missed something - at least in MCDK cells its occurrence cannot be predicted"*

We did not address in our study if the occurrence of the 3:1 motif is determined or not. We tested, if a parental cell determines the variability in cell cycle duration of its two daughter cells by analysing if a outlier cell of the 3:1 motif can somehow propagate its specialty in cell cycle duration to the next generation. Although determinism is found, we lack resolution on the question, how.

- b) *"L-cells have the same empirically determined probability to divide to give rise to 0/0 or L/0 cells"*

No, the empirical probabilities that L-cells divide to give rise to 0/0 or L/0 cells are not the same. This becomes visible in Figure 2c and 2d in the current manuscript, pointing to the importance of keeping this data in a main figure. Figure 2c shows that only one additional division of the type 0 to 0/0 would need to be found empirically in the data set to state that this division type is equally frequent as random. In a more abstract form Figure 2d shows the same namely, the empirical probability (0.4) is close to the peak of random frequencies (0.5) and thus clearly not significant different from random. In contrast, for the division type 0 to L/0 it is visible in Figure 2c that at least 3 additional empirical divisions of this kind would need to be found empirically to state that it is random for the specific

depicted randomization situation. Relative to 10'000 randomisations its frequency of 0.94 is very close to the upper confidence limit of 0.975 and thus nearly significant (Figure 2d).

c) The above reasoning of the referee is correct and prompted us to an important analysis.

A colony founding cell (having been a part of a lineage tree) could indeed give rise to two daughter cells that differ strongly, one being an outlier cell, the other not. Also correct is the reference to the bootstrapped analysis of daughter-daughter pairs that suggests that these are relative to the other comparisons alike (median Spearman coefficient about 0.6) (Figure S3a in the current manuscript). Cell cycle duration variation is relatively low but present between daughter-daughter cells. Thus, the question arises, how to reconcile a relative low variation of cycle duration and the possibility that outlier cells could be present in generation 2.

To address this, we analysed per generation the correlations of cell cycle durations of cells of individual daughter-daughter pairs (empirical) (Figure S14 in the current manuscript). Indeed, supporting the thought extrapolation, in generation 2 cell pairs exist that have very divergent cell cycle durations in the si non-target condition; such pairs could be composed of outlier and non-outlier cells. A similar result is also obtained for all following generations. Also consistent with conclusions of our study is the finding that the frequency of divergent cell cycle durations in these daughter-daughter pairs is higher in the si non-target condition than in the si ninein condition. Thus, satisfactorily, our new data (Figure S14 in the current manuscript) reveals that sibling cells (daughter-daughter pairs) in generation 2 can indeed strongly differ in cell cycle duration supporting the thought deduction and further suggesting the presence of a division asymmetry of some colony founder cells generating an outlier and non-outlier cell in generation 2.

The cell cycle durations in si non-target and si ninein MDCK cells decrease with increasing generation ranks (Figure S13b in the current manuscript). We observe such a trend also in cultures of mouse embryonic stem cells and mouse haematopoietic stem cells (Figure S13d, e in the current manuscript). Yet, it is not present when analysing *E. coli* or *C. elegans* lineages (Figure S13c,f in the current manuscript). We do not know the reason for these phenomena. Yet importantly, there is no correlation between the trend of cell cycle duration per generation and the result of the propagation analyses. Also, the outlier motifs do not change significantly over generations (Figure S3d).

The C. elegans data is interesting, but remains a correlation. We know that these lineages are non-random (a series of well characterized asymmetric divisions leads to the generation of the germ line). One can even find the longer cell cycle lengths of the cells in the P lineage in worm book. So, is it surprising that this lineage follows a pattern as found here? These divisions are also asymmetric in size (The P0 cell is smaller than the AB cell for instance, and there is a clear effect on cell size and the length of the cell cycle (e.g. Arata et al Front. Physiol. 2015). So, cell cycle length is perhaps a consequence of cell size and more factors than cell cycle length are involved in specifying this lineage and its potential. In the P lineage, it appears that cell cycle length can serve as a predictor for another division producing cell cycle length variations, which is also different to MDCK cells. Perhaps a bit more cautious discussion is needed here.

Reply: We thank the referee for the idea of a correlation with cell size. This is a valuable input and also mentioned by Referee 3. Therefore, we incorporated it in the modified discussion section of the revised manuscript.

The novelty in the *C. elegans* analysis is that the cells that are indeed known to have long cell cycle durations belong to the here identified 3:1 motif.

In the current manuscript we restructured the argumentation motivating better the use of this and further model systems.

Minor

Summary:

“its” third line– not clear what this refers to“

Reply: We write now: “As cell cycle duration varies amongst cells in mammalian tissue culture cells, we asked whether their division asymmetry contributes to this variability.”

.but appeared to emerge from asymmetric divisions taking place at non-random levels: a parent cell determines the unequal partitioning of the cell cycle duration in its two progenies”. Can you really say that, as there is no way to predict which parental cell will do it? What is “levels” referring to, this would be important to elaborate in the discussion.

Reply: The statement is correct, as the entire propagation analysis with si non-target cells was significantly different from random with a 0.005 error probability (see Figure 2d,e).

“..but appeared to emerge from asymmetric divisions taking place at non-random levels”

Reply: The term “non-random levels” refers to the significance level chosen in our statistical propagation analysis composed of division types containing outlier cells. It was here - as in most standard statistical tests - 5%, with 2.5% on each side of the distribution. Thus, for any result below 2.5% or above 97.5% the null hypothesis, stating that there is no propagation, is rejected.

I think the figure legends could generally use some work to clarify, what is shown.

Reply: We altered the figure legends giving more details.

Fig 1C:

This excludes cell1 and cell 1112? Correct? Please clarify in legend.

Reply: Yes, this is correct. We revised the legend text in the current manuscript to differentiate now clearly between imaged cell lifetime and cell cycle duration, which can only be determined if a complete cell lifetime is imaged, i.e. from cell “birth” to its next division.

Fig S7:

“Representative immunofluorescence images of si non-target and si ninein cells. The cells labelled for ninein (green), γ -tubulin (red) and nuclei (blue) show a lack of centrosomal staining in depleted cells at 0 h (94% +/- 0.88%) and 24 h (80.7% +/- 1.76%) but not at 100 h (5.1% +/- 3.18%) after imaging started.” C shows 24h, 48h, 124h not 0h, 24h, 100h, so what is it?

Reply: We apologize for the confusion and altered the figure and legend to increase clarity (Figure S1 in the current manuscript). The mismatch in numbers was caused by different reference time points in the figure and the legend text.

If the duration of the si ninein effect wears dramatically off at ~100/124h, and the imaging goes for 85h it may be good to include 90h of si ninein as a control, showing that the depletion is still efficient.

Reply: We point to the mRNA analysis at the 48 h time point in Figure S1b in the current manuscript, where the downregulation is still strongly detectable. Additionally, in the Figures SF3d, or SF13 at generation 5 and generation 6, si ninein and si non-target readouts differ showing that the cells in the different conditions differed in these generations. As the corresponding time points are in the range of 70 h to 95 h after cell seeding, the si ninein effect is still detectable at 90h.

Answers to the comments made by Referee 3.

It is known that in overtly asymmetrically dividing cells, a variation in cell cycle duration is seen between the two daughters. This paper asks whether a similar asymmetry between daughters is established at the time of division even in lineage where cell division is apparently symmetrical. In this way the authors seek to identify a cryptic asymmetry through careful mathematical analysis of the temporal aspect of lineages. This is a very clever and interesting idea. I really like this paper, I think it is a valuable addition to the cell heterogeneity literature and will attract a broad readership. Its elegance reminds me of the work by the "phage group" of Delbrueck et al back in the early days of molecular biology.

The main approach is a computational search for cell division motifs in lineage data. The description of this analysis is a bit complicated, because the analysis IS complicated, but I feel that the authors did an excellent job in motivating the idea of what they are doing so that readers who don't want to fully delve into the details are able to still understand the general idea, and for those who do want to delve into the details, I felt that the procedure was carefully described such that it would indeed be possible to replicate it based solely on this description.

Reply: We thank the referee for her/his positive opinion and support for our approach and style to present the study.

The inclusion of ninein RNAi as a control is extremely important for several reasons. First of all it gives insight into potential cell biological bases for the asymmetry, but I think even more importantly, with what is essentially a pattern detection algorithm, the question is always how likely would such patterns be accidentally detected when in fact there is nothing there to detect, i.e. what is the false positive rate for detecting a certain type of motif. There are various ways to try to generate synthetic data or otherwise estimate such errors, but it is always far more convincing to be able to provide a negative control data set, and then analyze that for comparison.

Reply: We agree with the referee and restructured the flow of argumentation. In the current manuscript we present the si ninein data set as a biological control to the si non-target data set and go further with presenting the analysis of additional cell systems to eliminate the possibility that si treated MDCK data represents a peculiarity, as it is a unique cell system.

The authors did a good job of exploring several possible bases for the 3L motif, ruling out variation in mitotic timing or position in the clone. However, they do not appear to have investigated cell size, or at least I could not find any place where this was discussed. The model that I have in mind is that perhaps a cell is born that is unusually small. The mechanisms that regulate cell size are, of course, not really known, but in general there is the idea that cells may grow to a defined size before they are able to divide. Such a "sizer" model for cell size regulation would predict that cells born small would have a longer interphase before their next division. By the same token, if the 3L pattern is not accompanied by variation in at-birth cell size, it would be interesting to know if the cells which grow longer end up larger, because they have had more time to accumulate biomass. I feel that even a simple quantification of at-birth and at-division cell size might be of great interest to the cell size community and could reveal new insights into this question, possibly helping to resolve the ongoing debates about sizers versus timers versus adders. This is just a suggestion, I wouldn't want to force the authors to do this, but I think it might be extremely interesting.

Reply: We agree with the referee's suggestion that was also put forward by Referee 2. But studying cell size was neither possible with the available MDCK data set nor with that of any of the other cell systems. Entering in this area would require a complete new study. As we also envision such a study promising we proposed it in the discussion section of the current manuscript.

I am confused by the transition rate analysis of figure 3. I would have thought that the rate of O/L should be balanced by the rate of L/O, since asymmetric transitions would suggest that over time more and more 3,L motifs should be occurring. But this could well be a confusion on my part as to how the transition matrix is constructed. I am interpreting the matrix as a standard Markov transition matrix, which would be multiplied by the distribution of motifs at a given generation to find the distribution at the next generation. I think perhaps the matrix given here is actually the product of such a Markov transition matrix with the distribution vector, i.e. the matrix entries are weighted by the probability of a given initial motif. I am not familiar with the term “ empirical transition matrix” but I am getting the impression that this is what that term describes. I would therefore suggest that the authors might want to be a little bit more explicit about what these matrix entries mean. i.e., they do not mean the probability that a cell from a given motif will produce progeny showing another motif, but rather the probability that a cell chosen from a population is part of a certain motif AND that it then produces progeny that are part of a certain motif. So that would explain why there are so many more O/L than L/O, simply because there are more O than L in the population. But then I do wonder, if the authors were to correct for the abundance of the initial motifs, do they now see an equal conditional probability that a cell from a O motif produces L daughters as that a cell from an L motif produces O motif daughters? I Otherwise, I would be curious to know whether there is a different overall abundance of L motifs in older colonies.

Reply: We regret to cause confusion and try to clarify in the following. Our propagation matrix is not a standard Markov transition matrix. There are two major differences: 1. Our division type matrix is not normalized (i.e. Figure 2b in the current manuscript shows the absolute number of division types found in the complete si non-target data set and no probabilities). 2. In a Markov transition matrix **one** initial state is converted to **one** outcome state. However here one initial state (“from” cell class) is “converted” (linked with) into two daughter cells each with a specific class and not only **one** cell class: “From” **one** mother cell class links “To” **two** daughter cell classes. As “From” is a unary initial state and “To” a binary outcome state a standard Markov chain cannot be applied.

Additionally, our initial states are linked to the 4-cell motif that dictates the distribution of the cell classes in the lineage trees. We tried to illustrate this in Figure 2a in the current manuscript to facilitate the understanding of the “empirical division type matrix”. Together, our propagation test is not predictive for events over time, it assesses if the empirical frequency of division types (composed of S- and L- outlier cells) found in lineage trees is random or not.

The term “empirical division type matrix” simply meant that the absolute number of a given division type in pooled lineage data of a given condition is depicted. For example, the division type S to O/O was found twice and the division type O to O/O 43 times in the si non-target MDCK data set. We altered the text to clarify the description.

In Figure S3d in the current manuscript we demonstrate that there is no significant change of the diversity motifs over the generations. Thus, older colonies, i.e. higher generation number, do not have significantly more outlier cells than young colonies, i.e. low generation number.

With this we strongly hope to have resolved unclear aspects of the analysis. Moreover, we changed the terminology from transition matrix (which might have caused the confusion with a Markov transition matrix) to division type matrix.

The results with C. elegans are potentially complicated by the fact that each cell in the lineage is known to be undergoing a different fate decision than the others, i.e. it is not a homogenous background. So personally I find the MDCK results much more interesting. But there is nothing wrong with the C. elegans data.

Reply: We followed as outlined above the strict logic approach of statistics and present now “other cell systems” as controls for the si MDCK data. The results obtained with the other cell systems validates the MDCK propagation analysis result and we therefore exclude the possibility that the MDCK results are peculiarities.

Very small comment:

1. The patterns are described as “motives” in some places, and “motifs” in others. I believe that the word “motive” should be replaced with “motif”.

Reply: We apologize for our inconsistency in the previous submission. As explained in our reply to Referee 1, we use in the current manuscript the term “motif”.

References:

- 1 Homem, C. C. F. *et al.* Long-Term Live Cell Imaging and Automated 4D Analysis of Drosophila Neuroblast Lineages. *PLOS ONE* **8**, e79588, (2013).
- 2 Snow, M. H. L. Gastrulation in the mouse: Growth and regionalization of the epiblast. *Journal of Embryology and Experimental Morphology* **42**, 293, (1977).
- 3 Calegari, F. *et al.* Selective Lengthening of the Cell Cycle in the Neurogenic Subpopulation of Neural Progenitor Cells during Mouse Brain Development. *J Neurosci* **25**, 6533-6538, (2005).
- 4 Arai, Y. *et al.* Neural stem and progenitor cells shorten S-phase on commitment to neuron production. *Nature Communications* **2**, 154, (2011).
- 5 Schnabel, R. *et al.* Assessing normal embryogenesis in Caenorhabditis elegans using a 4D microscope: variability of development and regional specification. *Developmental Biology* **184**, 234-265, (1997).
- 6 Clark, M. W. *et al.* Periplasmic Acid Stress Increases Cell Division Asymmetry (Polar Aging) of Escherichia coli. *PLoS ONE* **10**, e0144650, (2015).
- 7 Filipczyk, A. *et al.* Network plasticity of pluripotency transcription factors in embryonic stem cells. *Nature Cell Biology* **17**, 1235-1246, (2015).
- 8 Hoppe, P. S. *et al.* Early myeloid lineage choice is not initiated by random PU.1 to GATA1 protein ratios. *Nature* **535**, 299-302, (2016).
- 9 Pampaloni, F. *et al.* Tissue-culture light sheet fluorescence microscopy (TC-LSFM) allows long-term imaging of three-dimensional cell cultures under controlled conditions. *Integrative biology : quantitative biosciences from nano to macro* **6**, 988-998, (2014).
- 10 Graser, S. *et al.* Cep164, a novel centriole appendage protein required for primary cilium formation. *The Journal of Cell Biology* **179**, 321-330, (2007).

Appendix

Referee figure 1

Stronger differences in cell cycle durations of daughter-daughter pairs in the si non-target compared to si ninein MDCK lineages.

The cell cycle durations of all daughter-daughter pairs were analyzed per experiment in the lineages of the si non-target or si ninein data sets. Left: si non-target MDCK, right: si ninein MDCK. Top to bottom: five independent experiments (E1 to E5, where E stands for "experiment"). X- and y-axis, cell cycle durations of daughter 1 (D1) and daughter 2 (D2), respectively, whereby D1 has an equal or shorter cell cycle duration than D2. Red line: $x=y$; orange line: $x=1.25$ times y ; Red circles indicate daughter pairs above the orange line, i.e. the cell cycle durations of those daughter-daughter pairs differ by more than 25%. Red numbers in the plots represent the frequency of those daughter pairs whose cell cycle durations differ more than 25%; n, number of daughter-daughter pairs.

Referee figure 2

Example of short cell cycle duration assignment.

Time-lapse images of the si non-target lineage (ID 3), where a cell had a cell cycle duration of 3.5 h assigned. This cell of interest is marked with a red asterisk. Daughters of the cell of interest are marked with blue squares. Green box around the time points representing the cell cycle duration of the cell of interest. Time indications; time point after imaging start. Bar 5 μm .

Referee figure 3

Impact of short cell cycle durations on the statistical propagation test.

Division type matrices as in Figure 2f of the manuscript. The statistical propagation analysis was performed with the si non-target or si ninein MDCK datasets with the full data set as in the manuscript (top row), or with the 5 (middle row) or 10 shortest (bottom row) cell cycle durations eliminated from the data sets. Left: si non-target; right: si ninein MDCK. Procedure as described for Figure 2 of the manuscript; number of available and analyzed divisions indicated per division type; n, number of divisions with outlier and 0-cells.

Reviewers' comments:

Reviewer #2 (Remarks to the Author):

This manuscript describes a very interesting observation that is relevant to a broad readership. I appreciate the thorough response to the concerns raised previously and think that the authors have done a fine job in addressing them, from which the manuscript has clearly benefitted. The rationale for the use of Ninein siRNA could perhaps have been more elegantly introduced in the results section, but that is a minor issue. From my point of view there is no reason to delay publication.

Reviewer #4 (Remarks to the Author):

I had the pleasure to reevaluate the changes made to a manuscript by Berge et al. In particular, I was asked to judge whether the comments of two reviewers (1 and 3) have been adequately addressed.

While I share the skepticism of reviewer 1; I agree with reviewer 3 that the analysis of division events in pedigree structures is timely and interesting. While the subject is interesting, critical questions remain in place. From my point of view there are several, serious terminological and methodological issues.

On top of these, I am most worried about the conclusions drawn. This culminates in the authors' statement (l155) "Therefore, we concluded that determinism underlies the distribution of the division types in si non-target MDCK lineages." First, it would be necessary to define "determinism". As far as I understand there is no sign in the data to predict the outcome of any of the asymmetric events in advance. However, this is what determinism would imply, at least to my understanding. The only thing I can see is that certain motifs appear more pronounced than expected, indicating that there is a preference for some level of inheritance. It might well be the case that some divisions confer a higher tendency for longer cell cycle times, but I would avoid to call that "determinism". Moreover, the authors need to discuss not only significance but also relevance. How much are the proportions of the motifs shifted? Can this rule out "stochasticity" and confirm "determinism". The data is all but not black and white!

This raises the next critical point as the term "asymmetric division" is not well defined. Are we talking about asymmetric fate (in the sense the cell cycle durations of the daughters differ) or are we talking about functional asymmetry (such that the intended propagation of a difference between the daughters has a functional consequence)? The latter might be the case, and the authors aim to use "fate data" to validate this idea. However, they should take care to separate those different concepts (which fails already in the abstract).

While the few suggestions of the reviewer 3 were addressed adequately, I suppose that reviewer 1 is still not pleased with the current version of the manuscript.

Concerning the question regarding bimodality, the authors have now moved the corresponding figure to the supplement. The question whether this is a threshold phenomenon (distribution with some outliers) or a bimodal distribution (with distinct peaks) has not been answered. In fact, the graph in S3 is said to show two separate populations: pairs of cells that differ more and pairs of cells that differ less. But this information is (if at all) only indirectly reflected in this distribution, since it shows the distribution of subsampled correlation coefficients (each representing 60 cell pairs). Why don't the authors directly show the distribution of pairwise differences (e.g. the relative differences that are used to classify outlier pairs in referee figure 1 or fig S14)?

Furthermore, at least to my understanding, the method used is not bootstrapping (which would mean to draw a random sample of the same size as the original sample WITH replacement). The method used here is subsampling. I am doubting that the reasoning for using this method is correct. In their answer to reviewer 1 the authors state that Spearman correlation coefficients depend on the size of the sample, however it is the p value associated with the correlation coefficient that depends on the sample size, while the correlation itself remains unchanged.

I am also not at all convinced from Fig. 1c,d to show why the 3:L motif is overly abundant. In fact, the authors refer to some discordance between the observed data and some 95%CI. However, this is only true in a region between "2 and 3.5", if I understand the argument correctly. At least the lower threshold at dissimilarity index = 2 should be indicated, to make clear from which part of the plot their conclusion is actually drawn.

Furthermore, the request for a proper null model has not been sufficiently answered. While I agree that permutation tests are an appropriate tool to see whether observed correlations are more abundant than expected, I would suggest a computational study in which the authors can show (based on an underlying ground truth that generates pedigrees with particular features) that their method is uniquely able to recover those differences. Using such a modeling approach could also help to discuss the concept of "determinism".

I agree with reviewer 1 that the logic of the permutation tests is not well illustrated in (now) Figure two. Rearranging the subfigures is not enough to illustrate the concept behind.

There is a critical remark concerning the randomization strategy (Fig. S4): Could the significant overrepresentation of 3:L and underrepresentation of 3:S motifs be caused by differences in cell cycle durations between experiments or between cell lineages within experiments (cf. Fig S2)? Using the particular permutation strategy all cell cycle times of the same generation between all experiments are mixed, so could overall differences in the cct distributions impact the conclusions?

Following this line of thought, the results shown in Figure 2 could just reflect the overrepresentation of L cells and underrepresentation of S cells found in Figure 1 and cannot answer the question of whether there is an inheritance of motifs. Using the particular randomization strategy, one gets less L and more S cells after permuting cell cycle times. This then automatically leads to less $O \rightarrow L/O$ and more $O \rightarrow S/O$ transitions and could explain the "non-randomness" in Fig 2f. A proper randomization strategy to this question might be: Leave the sets of four granddaughter cells intact and reshuffle them completely between mother cells. This leaves the numbers of O, S and L cells constant, but randomizes the association of transitions.

The authors further argue that in the additional cell systems analyzed, the 3:L motif is repeatedly overrepresented if differentiation is involved. I don't see a display of this data. Although S13 contains stacked bar plots, it is not clear how they differ (again: more a question of relevance rather than significance) and how they compare to a null model.

The question regarding movies and pedigrees is not really answered. Is it "one movie, one pedigree?" Why is there only one pedigree per movie? Were they imaged on individual initial cells?

There are further, general and technical questions that remain to be answered:

- The description of the method to detect motifs in granddaughter pairs (Fig S4) appears to be much more complicated than necessary. In this one-dimensional case (just looking at cell cycle times) the Mahalanobis distances simply reduce to differences in cell cycle lengths and the hierarchical clustering is not needed at all. The motifs can just be determined based on where the largest difference is.

- I do not understand what is done in step 2D. Why can't you directly get the p values from the CDF in 2c?
- I also do not understand the idea of the power testing in Fig 2e.
- "Si ninein" in Figures 3 and S7: There are two p-values below 5% ($0 \rightarrow 0/0$ and $L \rightarrow 0/0$). Why are those not marked as significantly different? The same in E.coli 7.5 (Fig.s 4 and S8) and mESC (Fig.s 4 and S10). Also the numbers in the transition matrices in Fig.s 4B and S10 are not the same.

Response to reviewers

Reviewer figures and their legends are in the Appendix following this point-by-point response.

Reviewer #2 (Remarks to the Author):

This manuscript describes a very interesting observation that is relevant to a broad readership. I appreciate the thorough response to the concerns raised previously and think that that the authors have done a fine job in addressing them, from which the manuscript has clearly benefitted. The rationale for the use of Ninein siRNA could perhaps have been more elegantly introduced in the results section, but that is a minor issue. From my point of view there is no reason to delay publication.

We appreciate the positive reception of our changes. Concerning the introduction of si ninein we wish to state that there are two strategies: one coming from a biological argumentation, where si ninein altered the fidelity of stem cell divisions (Wang et al; mentioned and cited by us) and a second one where si ninein is in principle only a control to the results obtained with si non-target and the biology behind it is for this manuscript irrelevant. We tried to merge both to highlight the logic-based approach for the surprising result we obtained, namely a higher variability in the si non-target condition compared to the si ninein condition. Thus, the si ninein data is still introduced as follows: “To investigate further the possibility that diversification is cell autonomous, we tested whether modulating the frequency of asymmetric cell divisions alters the pattern of outlier generation.”

Reviewer #4 (Remarks to the Author):

I had the pleasure to reevaluate the changes made to a manuscript by Berge et al. In particular, I was asked to judge whether the comments of two reviewers (1 and 3) have been adequately addressed.

While I share the skepticism of reviewer 1; I agree with reviewer 3 that the analysis of division events in pedigree structures is timely and interesting. While the subject is interesting, critical questions remain in place. From my point of view there are several, serious terminological and methodological issues.

We thank this reviewer for taking time to try to in detail understand our study and judge the comments of previous reviewers, a difficult task. While we acknowledge and appreciate the identification of some shortcomings (e.g. see below at 5), 6), 10) and 12.4)) we have the impression that some important aspects in our study were not fully intelligible. We therefore included several new illustrative figures into our manuscript (Figures 2b, S2, S7, S8, 3rd submission) that shall help the reader to better understand the concepts. Specifically, we point here to Figure S2 (3rd submission) illustrating important terms used in our study especially emphasizing that the results of our study are built on only relative outlying cell cycle durations amongst four granddaughter cells (and not absolute ones, as this reviewer seems to understand); an aspect emphasized also in the coloured cell cycle durations in Figure 2b (3rd submission). These relative outlying cell cycle durations can be detected and their outlier property sized by our outlier detection method (Figure S5, 3rd submission; Figure S4, 2nd submission).

We agree that it was a very difficult task for Reviewer 4 to evaluate the thoughts of Reviewer 1 but we would appreciate a balanced evaluation, as we think to have replied adequately to all the points Reviewer 1 had raised and as Reviewer 4 did not discover any major shortcomings in our approach.

In the following we will cite the comments of Reviewer 1 in italics, whenever we realize that the comment of Reviewer 4 refers to point of Reviewer 1 before we respond.

- 1) On top of these, I am most worried about the conclusions drawn. This culminates in the authors' statement (l155) "Therefore, we concluded that determinism underlies the distribution of the division types in si non-target MDCK lineages." First, it would be necessary to define "determinism". As far as I understand there is no sign in the data to predict the outcome of any of the asymmetric events in advance. However, this is what determinism would imply, at least to my understanding. The only thing I can see is that certain motifs appear more pronounced than expected, indicating that there is a preference for some level of inheritance. It might well be the case that some divisions confer a higher tendency for longer cell cycle times, but I would avoid to call that "determinism". Moreover, the authors need to discuss not only significance but also relevance. How much are the proportions of the motifs shifted? Can this rule out "stochasticity" and confirm "determinism". The data is all but not black and white!

The difficulty of our study is that it encompasses concepts and methods of multiple disciplines, in which one term has different meanings and habits of use, like the term "determinism".

Yet, to avoid further misunderstanding we eliminated the term "determinism" and clarify use of critical words with definitions. Specifically, we added the following text (3rd submission, result, p7):

"The following propagation test reveals if cell cycle durations of daughter cells are uniformly at random linked to mother cells, which we define here as a stochastic association, or if a non-uniformity exists, which we define as non-stochastic association between mother and the two daughter cells in terms of cell cycle duration."

We also altered the abstract (3rd submission) as follows:

"Remarkably, divisions involving outlier cells are not uniformly distributed in lineages, as shown by permutation analyses, but appeared to emerge from asymmetric divisions taking place at non-random levels: a parent cell influences with 95% confidence and 0.5% error the unequal partitioning of the cell cycle duration in its two progenies. Upon ninein downregulation, this variability propagation was lost and outlier frequency and variability in cell cycle durations in lineages were reduced. As external influences were not detectable, we propose that a cell-autonomous process, possibly involved in cell-specialization, influences cell cycle duration variability."

Predictability, shift of proportions of motifs, relevance of significant results:

The Reviewer's "relevance", which we interpret as magnitude of distance, or shift of empirical to random results can be directly read out from panels of the propagation test: a random frequency of one division type is within the 95% confidence interval in the CDF (cumulative distribution function) and non-random frequencies of division types lie outside this 95% confidence interval (see for details Reviewer Figure 1, Appendix, where we predict frequencies that are stochastic for the given permutation). See also Figure S8 (new in submission 3), where upper and lower limit values of the 95% confidence interval of the permutations are depicted. The expected value for a random scenario is around 0.5 in the CDF. Our analysis is even more generalized yet not translatable into absolute counts of division types in the permutation test (Figure 3d, 3rd submission; Figure 2d, 2nd submission). (For details on the permutations we refer to Figure 2b, 3rd submission).

- 2) This raises the next critical point as the term “asymmetric division” is not well defined. Are we talking about asymmetric fate (in the sense the cell cycle durations of the daughters differ) or are we talking about functional asymmetry (such that the intended propagation of a difference between the daughters has a functional consequence)? The latter might be the case, and the authors aim to use “fate data” to validate this idea. However, they should take care to separate those different concepts (which fails already in the abstract).

We carefully used the term “asymmetric division”, as we are aware of ambiguities associated with it. In general, we wrote explicitly to what it refers to like cell cycle duration.

Some specific examples of the

3rd submission:

- **Abstract:** “...whether their division asymmetry contributes to this variability...”
- **P3:** “...in specific aspects linked to cell fate ¹⁻⁴. However, whether such diversifying asymmetric cell divisions concern only specific cell types or are a more general process, i.e., whether apparently symmetrically dividing cells establish some level of heterogeneity, is unclear.”
- **End of p3, towards end of the introduction section:** “However, whether the variability of cell cycle durations observed during the clonal divisions of mammalian cells reflects some division asymmetry is unknown.”

In the 2nd submission these sentences read as follows:

- **Line 24/25 abstract:** “..Whether division asymmetry contributes to this variability....“
- **Line 42:** “..aspects linked to cell fate. However, whether diversifying asymmetric cell divisions concern only specific cell types or are a more general process....is unclear.”
- **Line 50:** ...asymmetric cell divisions giving rise to cells with specific cell fates...
- **Line 50-52 towards end of the introduction section:** “However, whether the variability of cell cycle duration, which is observed in many cell types, itself reflects some underlying asymmetry in cell division is not yet clear.”

The last examples (2nd and 3rd submission) given define which kind of asymmetry we analyse in this study.

This reviewer differentiates between functional asymmetry and fate asymmetry, a distinction not entirely clear to us. As biologists fate asymmetry is linked to a functional consequence of an asymmetric phenotype as a relative long cell cycle duration. For this example, one would ask if the cell with the long cell cycle duration has a specific additional phenotype or property e.g. stress resistance or harbouring the old centrosome or expressing proteins characteristic

of specific cell type e.g. neuron versus its progenitor cells. Such a question is evident and addressing it a necessary next step but not the scope to this study, as such an investigation is initially only correlative and not causative. In other words, it is not the scope of our study to clarify the function or fate of the outlier cells. Yet, in the discussion section (of the 3rd and 2nd submission) we named a few potential functional and fate correlations to be tested. Yet, this study establishes for the first time due to experimental analysis - and not by modelling - that there is **at all a non-stochastic difference between cell cycle duration of daughter cells and that cell divisions propagate this variability even in apparently symmetrically dividing cells** (symmetrically dividing in terms of fate of known cell type specifications).

- 3) While the few suggestions of the reviewer 3 were addresses adequately, I suppose that reviewer 1 is still not pleased with the current version of the manuscript.

We do not fully understand, where we did not respond satisfactorily to the concerns of Reviewer 1 with our 2nd submission.

- 4) Concerning the question regarding bimodality, the authors have now moved the corresponding figure to the supplement. The question whether this is a threshold phenomenon (distribution with some outliers) or a bimodal distribution (with distinct peaks) has not been answered. In fact, the graph in S3 is said to show two separate populations: pairs of cells that differ more and pairs of cells that differ less. But this information is (if at all) only indirectly reflected in this distribution, since it shows the distribution of subsampled correlation coefficients (each representing 60 cell pairs). Why don't the authors directly show the distribution of pairwise differences (e.g. the relative differences that are used to classify outlier pairs in referee figure 1 or fig S14)?

Original words of Reviewer 1:

Much of the later analysis of 'outlier cells' is based on the observation that some cells have a much longer cell cycle length than others. While this is not obvious in Fig. 1A, Fig. 2B suggests a bimodal distribution of pairwise cousin correlation coefficients. I would like to see if this bimodality is visible in 3 independant experiments.

We think that there might be a problem in the understanding – in fact also in the case of Reviewer 1. We do not “classify outlier pairs” (as mentioned above by Reviewer 4) in the Referee Figure 1 or Figure S14 of the 2nd submission as the outlier detection method was not applied to generate these data; in contrast, in these figures (Figure S17, 3rd submission; Figure S14 of the 2nd submission) only absolute differences in cell cycle duration are depicted.

Thus, we highlight in these figures only that cell pairs can differ to different degrees in their absolute cell cycle durations. We ourselves do not mention the term outlier when presenting these results in the manuscript. Thus, we want to clarify that the term “outlier” as the reviewer used it in the comment to us, is misleading, as not according to the use of it in our manuscript. In the manuscript we only mention “outlier” cells and “outlying” cell cycle durations after introducing our motifs and the analysis of these. Some of these motifs indeed contain outlier cells but to find them the outlier detection method (Figure S5, 3rd submission; Figure S4, 2nd submission) needs to be applied first. Further, our outliers, as we use the term, are only relative outliers within the group of 4 granddaughter cells and they are only introduced in the manuscript after the Spearman’s coefficient analysis was presented.

We also interpreted the concerns of the Reviewers 4 and 1 in such a way that they fear that the different populations we observe are a result of one population coming e.g. from experiment 1 and 3 and the other population from experiments 2 and 4. We analysed this aspect in the Reviewer figure 1 of the 2nd submission. We found that daughter-daughter pairs that differ more and some that differ less in all experiments. Thus, in each experiment this “bimodality” (a word not used by us in the manuscript) is indeed in absolute cell cycle duration differences detectable.

We provide here in the Appendix the suggested figure (Reviewer Figure 2). Analysing directly the absolute cell cycle durations of cell pairs reveals that there are clear differences in si non-target and si ninein conditions. We find also using this approach:

- a) cell pairs that differ less and cell pairs that differ stronger (si non-target, si ninein),
and
- b) in the si ninein condition less cell pairs are present that differ stronger compared to the si non-target condition.

But we want to highlight again that - throughout the manuscript - we do not refer to absolute differences in cell cycle duration but to relative differences within a group of four granddaughter cells. Correspondingly, we used whenever suitable the term “relative outliers” instead of “outliers”.

Further, we maintain the subsampled comparative Spearman’s correlation coefficient analysis (Figure 1c, 3rd submission; Figure S3a, 2nd submission) to embed our study in recent work and to illustrate also in our MDCK system that there is indeed a higher degree of variability across generations (mother daughter, si non-target) compared to that within generations. Further and importantly, this variability across generations and even within generations is reduced in the si ninein data set. Yet, we wish to clarify that the histogram showing the “bimodality” (Figure S3 b of the 2nd submission) is not relevant for our main results. Therefore, we do no longer display the histogram in the manuscript.

Moreover, the choice of “subsampling 60 cell pairs” was to be able to compare two conditions, namely si non-target with si ninein. As these two conditions contain different numbers of elements, the direct comparison (without subsampling to equilibrate the element numbers) would result in different confidences of the Spearman correlations (e.g. if one compares a correlation analysis based on 200 cells with one based 100 cells, the latter is a less confident the correlation result). This argumentation was already in our response to Reviewer 1. Thus, indeed we could show the data not subsampled but then the two conditions cannot be compared.

- 5) Furthermore, at least to my understanding, the method used is not bootstrapping (which would mean to draw a random sample of the same size as the original sample WITH replacement). The method used here is subsampling. I am doubting that the reasoning for using this method is correct. In their answer to reviewer 1 the authors state that Spearman correlation coefficients depend on the size of the sample, however it is the p value associated with the correlation coefficient that depends on the sample size, while the correlation itself remains unchanged.

We really thank the reviewer for this critical remark. Indeed, we did not do bootstrapping as previously written but subsampling throughout our study. In the revised manuscript (3rd submission) we corrected this error (legend Figure 1c, 3rd submission). As mentioned above: the reason for subsampling is to be able to compare the results of the 2 conditions. Indeed, if only one condition were to be analysed the subsampling would not be needed.

- 6) I am also not at all convinced from Fig. 1c, d to show why the 3:L motif is overly abundant. In fact, the authors refer to some discordance between the observed data and some 95%CI. However, this is only true in a region between “2 and 3.5”, if I understand the argument correctly. At least the lower threshold at dissimilarity index = 2 should be indicated, to make clear from which part of the plot their conclusion is actually drawn.

We thank the reviewer for highlighting this unclear statement in our result part of the 2nd submission: “We found that the 3:L motif is the dominating diversity motif over all tested thresholds (threshold 2: 3:L motif 55.5%, 3:S 9.5%, 2:2 motif 17.5%; flat motif 17.5% Figure 1c, S3c). “

In the 3rd submission this reads now (p6):

“we found that the 3:L motif is the dominating diversity motif over all tested thresholds (threshold 2: 3:L motif 55.5%, 3:S 9.5%, 2:2 motif 17.5%; flat motif 17.5% Figures 2c (vertical

dashed line), S4a).” Thus, we had added a plot to the relevant figure. In that newly added plot only the empirical frequencies of the diversity motifs (3:L, 3:S and 2:2) at all tested thresholds are displayed (Figure 2c,d most left plot; 3rd submission). Thus, the cited statement above (2nd submission) does not refer to the motif-specific comparison between the empirical frequency of a motif of interest at a given threshold to the permutation results at that threshold (inside or outside the estimated confidence intervals).

We newly indicated the thresholds of 2 as vertical dashed lines in the individual plots of Figure 2c, d (3rd submission; not present in Figure 1c, d, 2nd submission).

- 7) Furthermore, the request for a proper null model has not been sufficiently answered. While I agree that permutation tests are an appropriate tool to see whether observed correlations are more abundant than expected, I would suggest a computational study in which the authors can show (based on an underlying ground truth that generates pedigrees with particular features) that their method is uniquely able to recover those differences. Using such a modeling approach could also help to discuss the concept of “determinism”.

Original words of Reviewer1: - In the tree data, it is difficult to define a proper null model to compare the observations with what one would expect. The assumption that 'cell cycle durations are uniform' is extremely naive though. What if one draws from the observed distribution of cell cycle times? Maybe specifically for the generation the cousin cells observed are in. Would the number of 3L cousin groups be still higher than expected?

We had responded to the justified criticism of Reviewer 1 in our 2nd submission and provided new results in Figure 1c, d (2nd submission) that are on the basis of Reviewer 1's suggestion.

Indeed, we had chosen the permutation approach and a very specific one, as we want to test one specific working-hypothesis, namely: does the cell cycle duration of a mother cell have an impact on the cell cycle duration of its two daughter cells. Making a different permutation would not challenge our working hypothesis.

Our null model is: the mother to daughter associations are equally distributed.

The chosen permutation approach is the most simple one to test for this ancestry association. We agree that indeed more complex models could be generated using mechanistic models. For example, one could ask if the age of a mother cell impacts on the cell cycle durations of its daughters. Although such implementations will be indeed very promising for the future they are currently stand-alone publications as the recent examples show: ^{1,2}.

We aim to write for an interdisciplinary readership and therefore formulated our null models especially intelligible for a biological readership. In our eyes we have always clearly stated our null models in this 3rd and also the 2nd submission (in the following parallel examples of the 3rd and 2nd submissions are given in the same sequence, otherwise as NEW marked):

3rd submission:

- *NEW - Result, End of p4:* “We wondered whether the variability in cell cycle duration of related cells originates from intrinsic processes segregating asymmetrically in divisions, our **working hypothesis**.”
- *Result, p6:* “...Also, unlike the 2:2 motif, the 3:L motif was for several tested thresholds above (threshold 1.5, 2, 2.5, 3, 3.5, 4, 7) and the 3:S motif below (threshold 1.5, 2, 2.5, 3) the 2.5 – 97.5% confidence interval of a random scenario according the **working hypothesis** (Figure 2c).”
- *Result, p6/7:* “...we asked if a cell intrinsic process or noise contributes to the generation of the 3:1 motif. To this end we probed whether a cell with an outlying cell cycle duration propagates some effect onto the cell cycle durations of its two daughter cells in the next generation thus affecting the 3:1 motif frequency in the next generation. To statistically test this possibility, we analysed the positions of the relative outlier cells in the lineages comparing them to all the other cells, termed 0-cells (all cells that are neither L- nor S-cells) (Figures 2a, S3a, S4c). The following propagation test reveals if cell cycle durations of daughter cells are uniformly at random linked to mother cells, which we define here as a stochastic association, or if a non-uniformity exists, which we define as non-stochastic association between mother and the two daughter cells in terms of cell cycle duration.”
- *Result p7:* “...we tested if the empirical frequencies of these division types are also very likely to occur after random permutations (non-parametric Monte Carlo permutations) of the cell cycle durations within each generation across all trees, or not (Figures 2b, 3c-d, S7). Additionally, we assessed the likelihood to obtain the empirical result by chance (type I error test; Figures 3e, S7, S8).”
- *Result, p8:* “To investigate further the possibility that diversification is cell autonomous, we tested whether modulating the frequency of asymmetric cell divisions alters the pattern of outlier generation.”
- *Result, p8:* “We thus asked if the statistical result that in si non-target MDCK lineages outlier division types occur at non-stochastic frequencies reflects some biological reality or is a false positive result. For this we tested statistically if a treatment that could perturb propagation fidelity across divisions abolishes the statistical non-stochasticity.”

- **Result p8:** “There is the possibility that extrinsic rather than cell-intrinsic influences cause the difference between the two experimental conditions. Thus, we probed if the statistical propagation difference between si non-target and si ninein lineages might be caused by external influences.”
- **Result p10:** “Next we probed, whether the 3:1 outlier motifs can only be detected and their frequency only be influenced in si treated MDCK cells. We therefore analysed several other cell systems.”

2nd submission:

- p.6 “we asked if a cell intrinsic process or noise contributes to the generation of the 3:1 motif. To this end we probed whether a cell with an outlying cell cycle duration propagates some effect onto the cell cycle durations of its two daughter cells in the next generation thus affecting the 3:1 motif frequency in the next generation”
- p. 7 “we tested if the empirical frequencies of these division types are also very likely to occur after random permutations (non-parametric Monte Carlo permutations) of the cell cycle durations within each generation across all trees, or not. Additionally, we assessed the likelihood to obtain the randomization result by chance (type I error test; Figure 2e).”
- p. 7: “To investigate further the possibility that diversification is cell autonomous, we tested whether modulating the frequency of asymmetric cell divisions alters the pattern of outlier generation.”
- p. 7 “We thus asked if the statistical result that in si non-target MDCK lineages outlier division types occur at non-random frequencies reflects some biological reality or is a false positive result. For this we tested statistically if a treatment that could perturb propagation fidelity across divisions abolishes the statistical determinism”
- p. 8 “There is the possibility that extrinsic rather than cell-intrinsic influences cause the difference between the two experimental conditions. Thus, we probed if the statistical propagation difference between si non-target and si ninein lineages might be caused by external influences.”
- p. 9 “Next we probed, whether the 3:1 outlier motifs can only be detected and their frequency only be influenced in si treated MDCK cells. We therefore analysed several other cell systems.”

Also: the comment by Reviewer 1 referred to our analysis as presented in the first submission where we estimated the confidence intervals by using uniform distributions. There we completely agreed that our null model was incorrect and we provided an analysis suggested by the reviewer in the revised manuscript (i.e. 2nd submission).

We believe a statistical analysis of experimental data that is not based on assumptions - as this one - is superior over any computational study that is based on assumptions.

I agree with reviewer 1 that the logic of the permutation tests is not well illustrated in (now) Figure two. Rearranging the subfigures is not enough to illustrate the concept behind.

Original words of Reviewer1: - Fig. 3 is mostly explaining the method to generate the permutation test. This should go to the supplements I suggest. The interesting part 3B lacks a legend, but also examples from the data that illustrate the under and over represented transitions.

We thank the reviewer for this remark. We hope our new illustrations in Figures 2b, S7 (3rd submission) help the readers to understand the underlying concepts.

- 8) There is a critical remark concerning the randomization strategy (Fig. S4): Could the significant overrepresentation of 3:L and underrepresentation of 3:S motifs be caused by differences in cell cycle durations between experiments or between cell lineages within experiments (cf. Fig S2)? Using the particular permutation strategy all cell cycle times of the same generation between all experiments are mixed, so could overall differences in the cct distributions impact the conclusions?

We wish to clarify Figure S4, 2nd submission (Figure S5, 3rd submission) illustrates our outlier detection method, in which permutations are not involved.

In this 3rd submission we included a figure part illustrating the permutation principle and its effect on outlier number and position in lineage trees (Figure 2b).

We assume the reviewer is concerned that Figure 1c, d of the 2nd submission (Figure 2c, d, 3rd submission) is due to the variabilities between experiments. Inter-experimental differences can formally not be excluded to have an impact on our results on the frequency of diversity motifs and propagation tests. Yet:

Firstly, we want to highlight that the experiments were performed in parallel in the same multi-well-chambered microscopy slide for the two conditions, si non-target and si ninein. Thus, if inter-experimental variations were indeed the main reason for our significant propagation test in the si non-target condition we should **not** have identified a difference between si non-target and si ninein, as the differences between experiments follow qualitatively the same trends in both conditions. Specifically:

- The cell cycle durations are in none of the si ninein experiments higher than in si non-target experiments (Figure S3, 3rd submission; Figure S2, 2nd submission; Reviewer Figure 3a, Appendix).
- All tested deciles in all experiments have the same or lower cell cycle duration in si ninein compared to si non-target (Reviewer Figure 3b, Appendix).

- In none of the experiments exceeds the differences of median cell cycle durations between the conditions two hours (Reviewer Figure 3b, Appendix).

Thus, none of our single experiments seems to be extremely different from the other.

Secondly, we even tested if the inter-experimental differences affect the empirical motif frequencies and their relationship to per generation permuted cell-cycle durations. We therefore performed the analysis (here only at the Dissimilarity index value of 2) as in Figure 1c,d, 2nd submission (Figure 2c,d, 3rd submission) - this time leaving out each time one experiment (Reviewer Figure 3c, Appendix). With this test we assessed which impact single experiments had on the pooled result (Figure 2c,d, S4a, 3rd submission; Figures 1c,d, S3c, 2nd submission). In Reviewer Figure 3c (Appendix) we generally see the same results as in the pooled dataset (Figure S4a, 3rd submission; Figure S3c, 2nd submission). Specifically, in the si non-target condition the 3:L motif remains in all these tests the most frequent one and above 50% and in all but 1 case (without E4) clearly above random levels. However in the si ninein condition the frequency of the 3:L motif is reduced below 50% and always in the random confidence interval. Similarly, the 3:S motif was after leaving out single experiments in the si non-target condition in all but one plot (without E4) clearly below random frequency but in the si ninein condition always within the confidence interval of random frequency. **Thus, we exclude that single experiments are the cause for the significant result for the 3:1 motifs in the propagation test in si non-target condition** (Figure 3f, 3rd submission; Figure 2f, 2nd submission).

Short note to our above analysis, where the results are not identical but only qualitatively similar to the “all data pooled” result: the reason for this is a reduction in the number of elements, which results in wider confidence intervals of the randomisation tests.

Thirdly, we also could imagine Reviewer 4 assumes e.g. the following scenario: all empirical 3:L motifs are found only in lineage trees of one experiment. Our analysis in Reviewer Figure 3c reveals that this is not the case.

- 9) Following this line of thought, the results shown in Figure 2 could just reflect the overrepresentation of L cells and underrepresentation of S cells found in Figure 1 and cannot answer the question of whether there is an inheritance of motifs. Using the particular randomization strategy, one gets less L and more S cells after permuting cell cycle times. This then automatically leads to less $0 \rightarrow L/0$ and more $0 \rightarrow S/0$ transitions and could explain the “non-randomness” in Fig 2f. A proper randomization strategy to this question might be: Leave the sets of four granddaughter cells intact and reshuffle them completely between mother cells. This leaves the numbers of 0, S and L cells constant, but randomizes the association of transitions.

To clarify, the permutations that underlie the results in Figure 1c of the 2nd submission (Figure 2c, 3rd submission) are the same as in Figure 2, 2nd submission (Figure 3, 3rd submission). The new Figure 2a, b (3rd submission) should clarify this.

In Figure 1c of the 2nd submission (Figure 2c, 3rd submission) is visible that the 3:L motif is at threshold 2 over- and the 3:S motif underrepresented compared to the permutations. Yes, indeed, these qualities are directly reflected in significant over- and underrepresentation of division types involving L- and S-cells, respectively. Yet it is not a necessity that e.g. the 0 to L/0 0 division type is affected. For example *per se* also the division type L to L/0 could have been overrepresented to match the overall underrepresentation of the 3:L motif in Figure 1c, 2nd submission (Figure 2c, 3rd submission).

Further, we think that there might be a misunderstanding. We never asked “whether there is an inheritance of motifs”.

Regarding “...one gets less L and more S cells after permuting cell cycle times. This then automatically leads to less $0 \rightarrow L/0$ and more $0 \rightarrow S/0$ transitions and could explain the “non-randomness”: This reasoning is wrong, as the assignment of S- and L-cells is only based on relative cell cycle duration differences within a granddaughter set. Thus, after permutations very different numbers of S- and L-cells can be present (Figure 2b, 3rd submission).

In summary, we hope that with a better understanding of the permutations, outlier detection method and propagation test the concerns of the reviewer could be removed.

10) The authors further argue that in the additional cell systems analyzed, the 3:L motif is repeatedly overrepresented if differentiation is involved. I don't see a display of this data. Although S13 contains stacked bar plots, it is not clear how they differ (again: more a question of relevance rather than significance) and how they compare to a null model.

We think the reviewer misunderstood us. We never “argue” that the 3:L motif is overrepresented if differentiation is involved. We wrote in the discussion section of our 2nd submission on p. 12 “For future studies, establishing correlations between the outlier cells and cell size, cell differentiation markers, ageing reporters (aggregated proteins, DNA damage markers), the cilium or division potential might be promising first steps to identify the underlying molecular mechanism(s)...”

We were particularly careful and only discuss the possibility that the 3:1 motif - not only the 3:L motif - could be analysed in future studies with regard to cell specialisation:

For example:

- in the abstract of the 3rd submission we write: “possibly involved in cell-specialization, influences cell cycle duration variability.” In the 2nd submission we had already written in the abstract: “we propose that a cell-autonomous process, possibly involved in cell-specialization...”
- p10 result section 3rd submission subtitle: **“Propagation of outlier cell cycle durations is modifiable and relates to cell specialisation”**

Yet, the comment of the reviewer let us realize a mistake in our discussion section and we are thankful for that. We had written on p12, 2nd submission: “...*The frequency of the 3:1 motif could represent a useful reporter for cell specialization events occurring in cellular diversification processes...*” We revise in the 3rd submission the starting sentence of this section to “The non-stochastic propagation of cell cycle duration variability (i.e. a non-stochastic propagation matrix composed of relative outlier cells)” instead of the “*The frequency...*” This change is necessary for the reasons mentioned in our response to point 9) of this Reviewer and does in our opinion also resolve the issues of the null-model and relevance as these were addressed before for the propagation analysis. Consequently, in this section, we had explicitly named the analysed and differentiating cell systems in which the outlier cells of the 3:1 motifs propagate across generations in a non-stochastic manner.

“I don't see a display of this data” We did not specifically highlight which cell system is one, in which differentiation occurs. As also the term “differentiation” is ambiguous, we rather describe in this 3rd submission, as we did in the 2nd submission, the detailed results on p13, discussion section as follows in the 3rd submission (2nd word exchanged, “determinism”, “also” removed

and figure citations are altered compared to 2nd submission): “The non-stochastic propagation of cell cycle duration variability (i.e. a non-stochastic propagation matrix composed of relative outlier cells) could represent a useful reporter for cell specialization events occurring in cellular diversification processes. At pH6, an external stress to *E. coli* cells, but not at pH7.5, non-stochastic propagation was detectable (Figure 5b). This suggests that the tested cell classes (L-, S-, O-cells) are linked to a stress response possibly allowing individual members of the population to cell-autonomously regulate the cell cycle duration of their progeny to support the survival of the species. This seems to be achieved by an increase of O to O/O and reduction of O to O/S divisions. We identified in *C. elegans* lineages the germline precursor cells as L-cells (Figure 5f). These cells are prospective founder cells of new organisms, documenting their high division and fate potential. Finally, whereas embryonic stem cells in non-differentiation medium - remaining in the self-renewal mode, where no differentiation takes place - did not propagate outlier cell cycle durations above random levels (Figure 5c), differentiating haematopoietic stem cells did propagate cell cycle variability at non-stochastic levels (Figure 5d).“

- 11) The question regarding movies and pedigrees is not really answered. Is it “one movie, one pedigree?” Why is there only one pedigree per movie? Were they imaged on individual initial cells?

We added in the new “Terminology” Figure S2a (3rd submission) the link between single cell-selection and movies and write in this 3rd submission:

- p4 “ In both conditions, single cells divided forming colonies during imaging. In movies documenting colony formation individual cells could be followed over time and their nuclei were tracked up to a maximum of seven generations (Figures S2, S3, Supplementary movies 1, 2). From the 36 movies acquired of cells with the si non-target treatment in 4 independent experiments, we generated lineages trees and analysed the cell cycle durations (Figures 1a, S2, see Methods and ²⁰). ..”
- p15 “MDCK widefield imaging Immediately after attachment, isolated single, sparsely seeded, low H2B-YFP-expressing cells were selected and multi-position imaging was performed with an inverted Zeiss 200M microscope...”

Thus, initially individual cells were selected and subsequently imaged. Consequently, one initial cell results in one movie and in one lineage tree.

We wrote in the 2nd submission:

- p. 4 “From the 36 movies acquired of cells with the si non-target treatment in 4 independent experiments, we generated lineages trees and analysed the cell cycle durations (Figures 1a, S2, see Methods” in the main text and
- p. 14 “Immediately after attachment, single, sparsely seeded, low H2B-YFP-expressing cells were selected and imaging was performed with an inverted Zeiss 200M microscope [...] in up to 120 positions...” in the methods part.

12) There are further, general and technical questions that remain to be answered:

12.1) The description of the method to detect motifs in granddaughter pairs (Fig S4) appears to be much more complicated than necessary. In this one-dimensional case (just looking at cell cycle times) the Mahalanobis distances simply reduce to differences in cell cycle lengths and the hierarchical clustering is not needed at all. The motifs can just be determined based on where the largest difference is.

“granddaughter pairs” are not analysed in Figure S4 of the 2nd submission (Figure S5, 3rd submission). We analyse granddaughter sets, i.e. cell cycle durations of the 4 granddaughter cells.

We thank the reviewer for the intention to simplify our method, but there seems to be a misunderstanding. We searched for an algorithm to specifically detect 3:L, 3:S and 2:2 motifs, diversity motifs, as defined in our text:

3rd submission, p5:“... The theoretically most minimalist diversity motifs amongst the cell cycle durations of four cells are the following two: I) a 3:1 motif, where one of the cells has a very different cell cycle duration (outlier cell) relative to the other 3 and II) a 2:2 motif, where the cell cycle durations are more similar within a pair than between the pairs (Figure 2a). To capture in our lineages such motifs, that are based on relative differences within a granddaughter set we used the outlier detection method (ODM, Figure S5). In case the four cells of a granddaughter set cannot be separated into two clusters with a given threshold, the method assigns the granddaughter set a flat motif.... The 3:1 motif occurs in two manifestations: i) the **3:L** motif, one of the cells has a long cell cycle duration (L-cells) relative to the other 3; ii) **3:S** motif: one of the cells has a shorter cell cycle duration (S-cells) relative to the other 3...”

2nd submission, p5:“...The theoretically most minimalist diversity motifs amongst the cell cycle durations of four cells are the following two: I) a 3:1 motif, where one of the cells has a very different cell cycle duration (outlier cell) relative to the other 3 and II) a 2:2 motif, where the cell cycle durations are more similar *within* a pair than *between* the pairs..... The 3:1 motif occurs in two manifestations (Figure 1c inlets): i) the **3:L** motif, one of the cells has a long cell cycle duration (L-cells) relative to the other 3; ii) **3:S** motif: one of the cells has a shorter cell cycle duration (S-cells) relative to the other 3.”

With this definition, the following proposition of the reviewer: “The motifs can just be determined based on where the largest difference is.” is not adequate, as the variability amongst all 4 cell cycle durations needs to be considered and not the absolute biggest difference – please see also Figure 2b, 3rd submission.

We add here an additional example: If we had used the largest difference in the most left data set of Figure S5, 3rd submission (Figure S4 of the 2nd submission), then step 2 would reveal the cell1 and cell4 as such a pair (MD 2.236). But we do not know how their difference is relative to the differences to other cell pairs in this set of 4 cells e.g. cell2 relative to cell4. Thus, the hierarchical clustering step is necessarily needed. Further, introducing the dissimilarity index gives a readout where the magnitude of the relative differences amongst the 4 cells is expressible in quantitative terms, which allows the introduction of a threshold (expressing more or less sensitivity in the detection of diversity amongst the cell pairs in the granddaughter set).

Thus, the important aspect here is that we analyse relative differences in the sets of 4 granddaughter cells. We hope Figures 2b and S2 in this 3rd submission help to highlight this.

12.2) I do not understand what is done in step 2D. Why can't you directly get the p values from the CDF in 2c?

We hope to have clarified the steps of the propagation test with the new Figure S7 (3rd submission). Already in the 2nd resubmission, we wrote on p.7 “Additionally, we assessed the likelihood to obtain the randomization result by chance”, which is the direct answer to the question.

12.3) I also do not understand the idea of the power testing in Fig 2e.

We hope to have clarified the entire propagation test with the new Figure S7 (3rd submission)”. Further, already in the 2nd submission, we wrote on p. 28 regarding Figure 2e: “Frequency of significantly different division types based on permutations in d). Numbers in the graph represent the cumulative probability of having at least x significant division types (in d; x between 0 and 5). Here, 2 or more significant division types have a probability of 0.005.” This description is the answer to the question.

12.4) “Si ninein” in Figures 3 and S7: There are two p-values below 5% (O→O/O and L→O/O).

Why are those not marked as significantly different? The same in *E.coli* 7.5 (Fig.s 4 and S8) and mESC (Fig.s 4 and S10). Also the numbers in the transition matrices in Fig.s 4B and S10 are not the same.

We thank the reviewer for the critical eye and corrected the discrepancy of Figure 4 and S10, 2nd submission (Figures 5, S13, 3rd submission). However, as we display now (3rd submission) the previous absolute counts in the diverse empirical division matrices as relative frequencies this might no longer be evident. This new display allows easier comparison between the diverse model systems.

Concerning the other points (si ninein, *E.coli* 7.5) we mention here briefly, the two most critical aspects 1) two-tailed test boundaries need to be considered and 2) two criteria need to be fulfilled for a colour-code. We hope the Figure S7, 3rd submission, explains why the final si ninein division type matrix is not colour-coded. The same argumentation holds for those for *E.coli* pH7.5 (Figures 5b and S11, 3rd submission; Figures 4 and S8, 2nd submission) and in fact also for mESC (Figures 5c and S13, 3rd submission; Figures 4 and S10, 2nd submission).

References:

- 1 Sandler, O. *et al.* Lineage correlations of single cell division time as a probe of cell-cycle dynamics. *Nature* **519**, 468-471, (2015).
- 2 Strasser, M. K. *et al.* Lineage marker synchrony in hematopoietic genealogies refutes the PU.1/GATA1 toggle switch paradigm. *Nature Communications* **9**, 2697, (2018).

Appendix

Reviewer Figure legends

Reviewer Figure 1

“Relevance” or “shift” expressed as distance of empirical from expected values.

- A) One plot of Figure 3c, 3rd submission (Figure 2c, 2nd submission) is magnified to illustrate the shift of the empirical data point (red x) from the expected value at 0.5 for one division type. One can deduce that permutation frequencies between 6 and 14 of this division type corresponds to frequencies within the 95% confidence interval. Thus, an empirical frequency of 16 is clearly shifted away from the expected value of roughly 10.
- B) On the basis of the expected values of the permutation underlying the CFD plots in Figure 3c, 3rd submission (Figure 2c, 2nd submission), a division type matrix is depicted.

Reviewer Figure 2

Absolute cell cycle differences between daughter cells in two conditions.

In si ninein cells are less frequently high absolute cell cycle duration differences between daughter cells compared to si non-target cells.

- A) The absolute differences in cell cycle duration between all available daughter cell pairs is depicted for si ninein (left) and si non-target (right).
- B) The absolute differences in cell cycle duration between all cell pairs that are part of a granddaughter set (no necessarily daughter pairs) is depicted for si ninein (left) and si non-target (right).

The number of available and analysed daughter pairs is indicated (n). The dashed line indicates the 95th percentile of the distributions of the absolute differences in cell cycle duration of daughter cell pairs (A) or granddaughter cell pairs (B) of the si ninein condition. The same threshold in cell cycle duration difference was applied on the distributions of the si non-target cells.

The data show that there are more daughter and granddaughter cell pairs in si non-target that differ more compared to si ninein cells.

Of note: Absolute differences in cell cycle duration do not directly mirror relative differences in cell cycle durations, as they are determined in the motif determination, which were analysed in our manuscript. For instance, an absolute difference of 2 hours reflects 20% for two cells with cell cycle durations of 10 and 12 hours but only 6.7% for two cells with cell cycle durations of 30 and 32 hours.

Reviewer Figure 3

Variability between experiments.

- A) Boxplot representations of cell cycle durations per condition of all available data. Central mark – median (here: red); edges of horizontal box – 1st and 3rd quartiles; whiskers - 1.5 times the interquartile range (IQR). N, number of analysed cell cycle durations.
- B) Boxplot representations of cell cycle durations per experiment (up to 5, display as A)). No lineage tree was available in si non-target experiment 1.
- C) Influence of individual experiments on motif frequencies. Leaving out single experiments (E1 – E5, respectively): Frequency of the empirical (filled circle) diversity motifs (3:L, 3:S, 2:2) and the flat motif at dissimilarity index 2 in si non-target MDCK cells (upper row), and si ninein MDCK (lower row) relative to the 2.5% (lower) to 97.75% (upper) confidence interval borders (horizontal values of box), respectively, estimated by subsampling with per generation randomized cell cycle durations (10,000 permutations). (n= available granddaughter sets). * this condition is in fact identical to the all pooled data, as no lineage tree was available in si non-target E1.

Reviewer Figure 1

A) Division type 0 to 0/L (detail of Figure 2c, previous submission)

Blue circles represent the data points of the estimated cumulative distribution function of the frequency of the given division type.

Red X represents the empirical value of the si non-target MDCK data set: 16 times

B) one example of expected values for a of a stochastic result for the division matrix (includes rounding errors)

	to	S/0	0/0	L/0
from	S	0	2	0
	0	8	42	10
	L	1	5	1

Reviewer Figure 2

A)

B)

Reviewer Figure 3

A)

B)

C)

Reviewers' comments:

Reviewer #4 (Remarks to the Author):

In their revised version of the manuscript "Asymmetric division events promote variability in cell cycle duration in animal cells and E.coli" Berge et al. have greatly improved their explanation of the applied methodology. The added Figures are very detailed and now confer a good understanding of the steps that have actually been taken. The flow of the analysis is much clearer and technical details are conferred in the required depth and clarity. I also recognize that substantial changes in phrasing, especially the removal of the term "determinism", have put the paper in a better context.

I am still puzzled about the authors objection to report on the relevance/effect size of their findings. As with any statistical analysis that does not include an a priori sample size estimation, the statistical significance is just one aspect of the results. In fact one could increase the sample size (tracking more trees with more granddaughter sets) even further, and maybe other transitions (such as L -> L/O, which is not yet significant) also become significant. The relevance can only be determined in terms of effect size. Although Figure 3 provides the frequencies for the test case, it is very difficult to read out the corresponding value for the "average" null model from the CDF. It would be an easy task for the authors to state that the transition probability in the observed samples is xxx % higher/lower as expected by random. This would give the reader a much better understanding of how "big" the effects are. Even highly significant effects can be small!

I also appreciate that the authors defend their methodology. Still, it is not clear to me, why the authors focus on motifs among granddaughters to identify cells with relatively shorter or longer cell cycles. I am not saying it is not valid, but it is definitely not the most simple approach and would require some justification of its superiority. For example, the identification of short or long cell cycles could also be done by evaluating an overall distribution of cell cycle times and defining critical quantiles to identify outliers. Divisional asymmetry with respect to cell cycle duration of daughter cells can also be assessed by looking at the difference in cell cycle times and detecting deviations from the population average. This would allow to track the inheritance / non-inheritance of cell cycle durations. However, in the presented approach, the classification of a cell as being S, L and O is only relative to its grandsiblings. In fact, there could be S-cell with the same cell cycle length as some L-cells, only because the reference set is always different. It is necessary to show that such a relative definition (with respect to the grandsiblings) provides more insights as a global definition (with respect to the distribution of all cell cycle durations).

My initial concerns regarding the conclusions are still not ruled out. This is partly true to the methodological questions raised above, but in many parts it may also be a formulation problem that needs to be addressed. I point out some formulations that might be puzzling to a biological readership, although this list is not exhaustive:

- "These additional cell system results point further to a biological relevance for the L-, S-, and O-cell classes in diverse cell systems." As the authors stated earlier, there is a "association" with some cell cycle durations and their daughters (O do more L/O as expected). However, to derive a "biological relevance" usually requires to prove a functional difference (such as: these daughters preferentially differentiate to lineage x or undergo apoptosis). As such it is just an association between cells that were classified in comparison to their granddaughter-set and its offspring. Whether this is relevant in terms of function needs to be shown.

- "Here we identified parameters promoting the emergence of cell cycle duration variability in several clonally grown cell systems." I don't really see which parameter(s) have been identified. Beyond the statistical association, I can only appreciate that the downregulation of the centrosomal protein ninein disturbs this correlation. However, this is at most one parameter

which holds for one population. For E. coli the authors state that “molecular causes for the 3:1 motif differ”. I would just recommend caution with the word “parameters”, especially with its plural. Almost all results are associations and the functional/mechanistic part of how this “nonstochastic propagation” works are not deciphered.

- “The non-stochastic propagation of cell cycle duration variability (i.e. a non-stochastic propagation matrix composed of relative outlier cells) could represent a useful reporter for cell specialization events occurring in cellular diversification processes.” I agree with the authors that it would be useful if one could use the “ODM” to prospectively identify cells that will contribute to asymmetric divisions and confer certain fates to their daughters. But again, it is by far not deterministic (“every O-cell divides asymmetrically”) nor is there any functional correlation shown so far. In order to substantiate the above statement you would have to show that the O, S or L-type have something to do with function (i.e. different diversification among their offspring). The above statement is highly speculative and I do not see any strong evidence in the presented data.

Response

Alterations compared to 3rd submission highlighted followed by point by point response to reviewer.

I General alterations in 4th submission

Figures:

Figure 1b: robust z-score displayed in graph

Figure S7: 3 alterations as follows:

- Spelling in the word manuscript in text area upper right corner
- Number “0” was not aligned properly in most right graph in ii.1
- Behind the term “step iii. 1 a” ** were missing.

Tables:

- Addition of a new Table with legend, Table 2: Effect sizes for division types expressed as estimated robust z-scores. The legend in Supplementary information reads as follows:

*“Effect sizes reported as estimated robust z-scores are displayed for the individual division types in the indicated model systems and conditions, the relevant empirical number of divisions are reported in Table 1 (divisions with outlier and 0-cells (threshold 2)). For better orientation, colored fields repeat the schematic summary of the propagation test results of Figure 5a-e. NaN, not a number (when denominator was zero); *, rounded number.”*

Text:

Several sentences were added to be conform with the submission guidelines. All differences compared to the previous submission are highlighted as track-changes.

II Response to Reviewer #4 (in blue, with citations from the 4th submission in *italics*)

In their revised version of the manuscript “Asymmetric division events promote variability in cell cycle duration in animal cells and E.coli” Berge et al. have greatly improved their explanation of the applied methodology. The added Figures are very detailed and now confer a good understanding of the steps that have actually been taken. The flow of the analysis is much clearer and technical details are conferred in the required depth and clarity. I also recognize that substantial changes in phrasing, especially the removal of the term “determinism”, have put the paper in a better context.

I am still puzzled about the authors objection to report on the relevance/effect size of their findings. As with any statistical analysis that does not include an a priori sample size estimation, the statistical significance is just one aspect of the results. In fact one could increase the sample size (tracking more trees with more granddaughter sets) even further, and maybe other transitions (such as L -> L/0, which is not yet significant) also become significant. The relevance can only be determined in terms of effect size. Although Figure 3 provides the frequencies for the test case, it is very difficult to read out the corresponding value for the “average” null model from the CDF. It would be an easy task for the authors to state that the transition probability in the observed samples is xxx % higher/lower as expected by random. This would give the reader a much better understanding of how “big” the effects are. Even highly significant effects can be small!

We thank the reviewer for mentioning the term effect size, which we didn't recognize before as critical. Yet, we wish to point out that we had given already with the third submission in

Figure S8 an analysis of the size of the effect, as we reported the distance of the permuted probabilities to the empirical ones for the MDCK si non-target and si ninein data sets. Visible therein is for example that the empirical division type 0 to S/0 is 75% below of the expected value (2 versus 8, Figure S8 d versus b)). Similar, the empirical division type 0 to L/0 is 60% above the expected value (16 versus 10, Figure S8 d versus b)). In the new version of the manuscript we emphasize the result of this supplementary figure as well as the new table in an additional sentence. We write now in 4th submission result section, page 7 “*Additionally, we interpret the effect sizes of these shifts from random as biologically meaningful (SF8, Table 2).*”

Further, a reader can detect in each of the detailed figures of the propagation tests for each individual system and condition analyzed a panel reporting for each division type the estimated cumulative distribution function (CDF; Figures 3d, S9c, S11c, S12c, S13c, S14c). In this panel the complete distribution of the permutation results and the empirical value is displayed, allowing the assessment of the relevance of each empirical division type result relative to the permuted results.

Additionally, we added now Table 2, in which we report the effect sizes for the individual division types of the individual model systems as estimated robust z-scores, used for example in diverse biological screens (Birmingham et al., 2009; Lipinski et al., 2012; List et al., 2016). For example, a value of 3 means that the empirical value is three times the MAD (median absolute deviation of the estimated distribution) away from the expected value. This is the case for the division type 0 to S/0 and 0 to L/0 in the si non-target data set, the only division types of that data set that were in the propagation test significant. For this data set (MDCK si non-target) the value 3 for the estimated z-score was the maximal value in the division matrix fitting with the significance results. In general, in all analyzed systems and conditions the divisions that were tested significant had the maximal estimated z-scores of the given division matrix and these values range between 3 and 14.5. Thus, we interpret that our significant propagation tests go along with clear effects.

In this 4th submission we write in the methods section:

“Effect size

Effect sizes were expressed in two types of robust z-scores (rz-score). The robust z-score reported in Figure 1b was calculated according: $rz\text{-score} = \text{abs}(\text{median}(\text{si non-target}) - \text{median}(\text{si ninein})) / \text{avr}(\text{MAD}(\text{si non-target}), \text{MAD}(\text{si ninein}))$, with abs, absolute value; avr, average; MAD, median absolute deviation. “Si non-target” and “si ninein” refer here to the distribution of cell cycle durations of the mentioned condition. The pooled cell cycle durations of the two conditions differed by about 1 MAD. Thus, the effect size between these two conditions is in the same magnitude as half of all data points are distant from the median; which we interpreted as a clear effect.

The individual estimated rz-scores for the division types in the different model systems and conditions displayed in Table 2, reporting the absolute difference between the expected and the observed division type counts divided by the MAD of the expected count distribution, were calculated: $\text{estimated rz-score} = \text{abs}(\text{observed} - \text{median}(\text{expected count distribution})) / \text{MAD}(\text{expected count distribution})$, with the expected count distribution corresponding to the corresponding CDF-plots of the analysed division type (Figures 3d, S9c, S11c, S12c, S13c, S14c). In general, in all analysed systems and conditions the divisions types that were tested significant in the propagation test had the maximal estimated z-score values of the given division type matrix and these values ranged between 3 and 14.5. Thus, we interpreted that our significant propagation tests go along with clear effects. “

Finally, we also report the effect size of cell cycle duration distributions between si non-target and si ninein-treated MDCK cells (Figure 1b). Its magnitude is nearly 1 MAD. Thus, the effect size between those conditions is in the same magnitude as half of all data points are distant from a median; which we interpret as a clear effect.

In the method section in this 4th submission – also cited above - is the following text: “*The pooled cell cycle durations of the two conditions differed by about 1 MAD. Thus, the effect size between these two conditions is in the same magnitude as half of all data points are distant from the median; which we interpreted as a clear effect.*”

Table 2 mentioned in the 4th manuscript outside this method section as follows:

- P7 “...Additionally, we interpret the effect sizes of these shifts from random as biologically meaningful (SF8, Table 2).”
- P10 “..Figures 5b, S11, S12, estimated robust z-scores Table 2). “
- P10”... sets Figures 5d, S14, estimated robust z-scores in Table 2).”
- P10 “.0/0; 0 to L/0, L to 0/0; L to L/0) (estimated robust z-scores in Table 2).”

I also appreciate that the authors defend their methodology. Still, it is not clear to me, why the authors focus on motifs among granddaughters to identify cells with relatively shorter or longer cell cycles. I am not saying it is not valid, but it is definitely not the most simple approach and would require some justification of its superiority. For example, the identification of short or long cell cycles could also be done by evaluating an overall distribution of cell cycle times and defining critical quantiles to identify outliers. Divisional asymmetry with respect to cell cycle duration of daughter cells can also be assessed by looking at the difference in cell cycle times and detecting deviations from the population average. This would allow to track the inheritance / non-inheritance of cell cycle durations. However, in the presented approach, the classification of a cell as being S, L and 0 is only relative to its grandsiblings. In fact, there could be S-cell with the same cell cycle length as some L-cells, only because the reference set is always different. It is necessary to show that such a relative definition (with respect to the grandsiblings) provides more insights as a global definition (with respect to the distribution of all cell cycle durations).

The reviewer suggests to either define outlier cells based on “critical quantile” thresholds or to study “divisional asymmetry with respect to cell cycle duration of daughter cells” whereby the difference in cell cycle times would be classified as “deviations from the population average.” Thus, the classification should be between daughter cells and either based on “critical quantiles” or “deviations” from a “population average”.

Such approaches will not allow a stringent logic argumentation chain, as we present it in this study. For a better understanding of our subsequent argumentation we recapitulate here the steps of our study: 1) 3:1 motifs occur at non-random frequencies in non-target treated MDCK cells; 2) propagation test with non-target treated MDCK reveals that i) the position and frequency of outliers occur non-stochastically in lineage trees and ii) that divisional asymmetry with high likelihood exists; 3) documentation by propagation analysis of si ninein-treated MDCK reveals that this result is not false positive, 4) the comparison with other (non-MDCK) model systems is necessary to allow the statement that our outlier cells are not MDCK-specific and even occur in non-si-treated cell systems, where divisional asymmetry can also be observable. These details are listed to establish that focusing only on step 1 and 2) is not sufficient, but that especially steps 3) and 4) are mandatory for our conclusion.

In the following we list 1) caveats compared to our approach, where we define outliers based on the 4 closest related cells existing within one generation and 2) several arguments why the reviewer’s suggestions do not allow our stringent logic argumentation. Overall, there are two ways to categorise a continuous variable like cell cycle duration, i.e. to define S-, L-, and 0-cells based on cell cycle durations: a) by keeping the reference set (used for defining categories) constant (global approach) or b) by keeping the degree of relationship constant. The reviewer suggests to use the first, we opted for using the second for the following reasons:

1. A major problem is that the choice of the “critical quantiles” or “deviations” are not biologically motivated in the reviewer’s approach, in contrast to our ODM. Applying the ODM and the propagation test we study the question: **Are cells, that deviate more than double in their cell cycle duration compared to the biggest deviation amongst their 3 closest related cells, propagated randomly to their progeny?** Thus, in our ODM we determine and compare the diversity degrees amongst the 4 closest related cells and our threshold reports about a distance ratio of diversity amongst these 4 cells. The distance ratio value of 2 (threshold mainly used in our study) is justified as it tests the above-mentioned question. As we study the origin of diversity of cell cycle duration within lineage trees, this is an intelligible approach that uses for the classification a comprehensible diversity criterium.
Yet, following the proposition of the reviewer the question which exact quantile/deviation to choose for the analysis would arise. Thus, i) addressing this question seems to be an endless endeavour and ii) any quantile found that would result in a significant propagation test would not express any understandable diversity measure to get at the root of the diversity of cell cycle duration within a lineage tree.
2. The cell cycle durations shift systematically over the generations for MDCK (si non-target and si ninein), mESC, mHSC and *C. elegans* cell systems (see Figure SF16 b, d, e, f). Thus, classification with a global approach based on a population average would introduce a bias in the frequency of outlier cells (e.g. cells with extra-long cell cycles would be underrepresented in higher generations if a 10% cut off would be chosen) in individual lineages and would thus distort the propagation analysis. Moreover, the dependencies of cell cycle durations and generations differ in our tested model systems. Hence a comparison of the global-propagation tests would not be possible (step 3) and 4)) and the full argumentation chain disrupted.
3. As the population distributions of our different model systems vary in dispersion, skewness, and kurtosis, the definition of one “critical threshold” or ”deviation” would affect the different systems differently. Hence a comparison of the global approach-based propagation tests would not be possible (step 3) and 4)) and the full argumentation chain disrupted.
4. Information propagation is a function of time or in our case of generations. In a global approach, one would define those “critical quantile thresholds” based on data pooled over all generations, which has no sensible justification as over time changes are potentially introduced into the studied biological cell system. Further, justification is missing for the fact that in a global approach future properties (cell cycle durations in late generations e.g. generation 5) should be relevant for the outcome of e.g. a cell division in generation 2. Hence, a global approach would not be appropriate to analyse information propagation in lineage trees.
5. The two approaches differ in the questions underlying the classification. For the global approach the question is (the question of our study is formulated in bold in reason 1)): Are cells, that deviate more or less than a “critical threshold” or “deviation” of all (including early up to late generations) cell cycle durations of the lineage trees of a certain condition, propagated stochastically to their progeny. The biological relevance of such a question is doubted. Further, as the classification questions differ also the questions underlying the subsequent propagation tests differ. Thus, a completely different study would be generated employing the global approach.

In conclusion, a global approach to define outliers would i) study a different biological question than ours, ii) would violate the independence of variables, and iii) would not allow the comparison of different model systems, an essential aspect of our argumentation chain. Therefore, it is not sensible to employ such a global approach to our study.

Finally, we wish to share biological reasons that motivated our approach, being aware that these are *per se* not arguments against a global approach:

In biology important milestone results are obtained by the analysis of cells in the granddaughter generation or F2 as it is abbreviated in genetics. For example, the landmark experiments leading to Mendel's rules, the semiconservative DNA replication ¹, or centrosome maturation ² are based on analysing cells at the granddaughter level. In this context it is noteworthy that ninein localizes at unique appendages at the mature and not immature centriole. Si ninein treatment thus likely interferes with centrosome maturation. Also, in the cell cycle duration field it was recognized for decades that there is a special strong correlation amongst granddaughter cells in mammalian cells and bacteria ³⁻⁶, without an understanding of why. Our study now establishes a method with which this could be tackled in the future.

Together, although the global approach might seem simpler at the first sight, we highlight that it differs in the question studied and present logic (reasons1-5) and biological arguments justifying our more complicated approach to study our question.

In the 4th submission we now write explicitly in the method section after the ODM: "*Studying the link between information propagation and diversity of cell cycle duration we used a local in-generation reference (among 4 related granddaughter cells) to classify outlier cells allowing to robustly define and especially to compare outlier cells in diverse model systems despite 1) the variable dependencies of cell cycle duration on generations (Figure S16b-f), 2) their different moments in cell cycle duration distributions and 3) possible differential time dependent changes in cells of each system. Such a study is not possible employing a global approach (based on an entire population distribution i.e. cell cycle durations across several generations) for outlier classification.*"

My initial concerns regarding the conclusions are still not ruled out. This is partly true to the methodological questions raised above, but in many parts it may also be a formulation problem that needs to be addressed. I point out some formulations that might be puzzling to a biological readership, although this list is not exhaustive:

The reviewer cites in the following parts from the discussion section, the place where authors can speculate. We did this and indicated this either by choice of the tense of used verbs or by the verbs themselves.

- "These additional cell system results point further to a biological relevance for the L-, S-, and O-cell classes in diverse cell systems." As the authors stated earlier, there is a "association" with some cell cycle durations and their daughters (O do more L/O as expected). However, to derive a "biological relevance" usually requires to prove a functional difference (such as: these daughters preferentially differentiate to lineage x or undergo apoptosis). As such it is just an association between cells that were classified in comparison to their granddaughter-set and its offspring. Whether this is relevant in terms of function needs to be shown.

We had used the term "point" in the cited sentence to be a bit stronger than only "indicate", as L-, S- and O-cell classes might indeed be biologically relevant. We still hold on to this term, as the repetition of significant propagation tests – where we now also now report about the magnitude of the effect (Table 2) – strongly indicate that the phenomenon of significant division matrices does not exist by chance. Our strongest argument for the claim that result-repetition in other model systems is significant is that *E coli* had in one culture condition a significant propagation test but not in the second one. Therefore, there are in our study two model systems that each react in propagation tests similarly modifiable, which lets us claim that cells that build the division type matrix should have a biological relevance. Yet, we completely comply with the reviewer that the function needs to be elucidated and to do so the molecular basis of the phenomenon first needs to be deciphered as already previously mentioned in the discussion section.

- “Here we identified parameters promoting the emergence of cell cycle duration variability in several clonally grown cell systems.” I don’t really see which parameter(s) have been identified. Beyond the statistical association, I can only appreciate that the downregulation of the centrosomal protein ninein disturbs this correlation. However, this is at most one parameter which holds for one population. For E. coli the authors state that “molecular causes for the 3:1 motif differ”. I would just recommend caution with the word “parameters”, especially with its plural. Almost all results are associations and the functional/mechanistic part of how this “nonstochastic propagation” works are not deciphered.

We agree that the molecular basis for the phenomenon is not yet identified, as it is part of a future study identifying the causation of the phenomenon. Yet, we used a plural term of “parameter”, as si ninein treatment in MDCK abolishes the association and also a change of pH6.0 to pH7.5 eliminates the association. Thus, although system-specific, two entry points for manipulation of the non-stochastic propagation have been identified.

- “The non-stochastic propagation of cell cycle duration variability (i.e. a non-stochastic propagation matrix composed of relative outlier cells) could represent a useful reporter for cell specialization events occurring in cellular diversification processes.” I agree with the authors that it would be useful if one could use the “ODM” to prospectively identify cells that will contribute to asymmetric divisions and confer certain fates to their daughters. But again, it is by far not deterministic (“every 0-cell divides asymmetrically”) nor is there any functional correlation shown so far. In order to substantiate the above statement you would have to show that the 0, S or L-type have something to do with function (i.e. different diversification among their offspring). The above statement is highly speculative and I do not see any strong evidence in the presented data.

We had intentionally chosen the conditional mood form of the verb “can” in the cited sentence to indicate that we indeed speculate. Our cited statement is built on the deduction of a unifying property of the several systems and conditions in which we identified non-stochastic propagation of cell cycle duration. The scope of this study was not to address biological functions but to lay the basis for future studies. This study here demonstrates the fact that non-stochastic propagation of cell cycle duration exists, even in evolutionary distant cell systems, that it is experimental tunable, and that asymmetric cell division contributes to it.

References:

- 1 Meselson, M. *et al.* The replication of DNA in Escherichia coli *Proceedings of the National Academy of Sciences of the United States of America* **44**, 671-682, (1958).
- 2 Chrétien, D. *et al.* Reconstruction of the centrosome cycle from cryoelectron micrographs. *J. Struct. Biol.* **120**, 117-133, (1997).
- 3 Froese, G. The distribution and interdependence of generation times of HeLa cells. *Exp Cell Res* **35**, 415-419, (1964).
- 4 Powell, E. O. Some features of the generation times of individual bacteria. *Biometrika* **42**, 16-44, (1955).
- 5 Staudte, R. G. *et al.* Additive models for dependent cell populations. *Journal of Theoretical Biology* **109**, 127-146, (1984).
- 6 Sandler, O. *et al.* Lineage correlations of single cell division time as a probe of cell-cycle dynamics. *Nature* **519**, 468-471, (2015).